# Learning High-Precision Bounding Box for Rotated Object Detection via Kullback-Leibler Divergence

**Xue Yang[1],\* Xiaojiang Yang[1], Jirui Yang[2], Qi Ming[3], Wentao Wang[1], Qi Tian[4], Junchi Yan[1]†**

[1]Department of Computer Science and Engineering,
MoE Key Lab of Artificial Intelligence, AI Institute, Shanghai Jiao Tong University
[2]University of Chinese Academy of Sciences
[3]Beijing Institute of Technology    [4]Huawei Inc.
{yangxue-2019-sjtu, yangxiaojiang, wwt117, yanjunchi}@sjtu.edu.cn
{yangjirui123, chaser.ming}@gmail.com   tian.qi1@huawei.com

## Abstract

Existing rotated object detectors are mostly inherited from the horizontal detection paradigm, as the latter has evolved into a well-developed area. However, these detectors are difficult to perform prominently in high-precision detection due to the limitation of current regression loss design, especially for objects with large aspect ratios. Taking the perspective that horizontal detection is a special case for rotated object detection, in this paper, we are motivated to change the design of rotation regression loss from induction paradigm to deduction methodology, in terms of the relation between rotation and horizontal detection. We show that one essential challenge is how to modulate the coupled parameters in the rotation regression loss, as such the estimated parameters can influence to each other during the dynamic joint optimization, in an adaptive and synergetic way. Specifically, we first convert the rotated bounding box into a 2-D Gaussian distribution, and then calculate the Kullback-Leibler Divergence (KLD) between the Gaussian distributions as the regression loss. By analyzing the gradient of each parameter, we show that KLD (and its derivatives) can dynamically adjust the parameter gradients according to the characteristics of the object. For instance, it will adjust the importance (gradient weight) of the angle parameter according to the aspect ratio. This mechanism can be vital for high-precision detection as a slight angle error would cause a serious accuracy drop for large aspect ratios objects. More importantly, we have proved that KLD is scale invariant. We further show that the KLD loss can be degenerated into the popular $l_n$-norm loss for horizontal detection. Experimental results on seven datasets using different detectors show its consistent superiority, and codes are available at https://github.com/yangxue0827/RotationDetection.

## 1 Introduction

As a fundamental building block for visual analysis across aerial images, scene text etc., rotated object detection has recently been developed rapidly [1, 2, 3, 4, 5, 6], which benefit themselves from the well-established horizontal detection approaches [7, 8, 9, 10, 11]. Specifically, many works [12, 13, 14, 15] build themselves upon the previously established horizontal detection pipeline from an inductive perspective, as shown in Figure 1(a). However, these detectors are often unable to cope with challenging scenes well due to the limitations of current regression loss, such as large aspect ratio objects, dense scenes, etc., resulting in obvious disadvantages in high-precision detection.

---

*Part of the work was done during an internship at Huawei Inc.
†Correspondence author is Junchi Yan.

35th Conference on Neural Information Processing Systems (NeurIPS 2021).

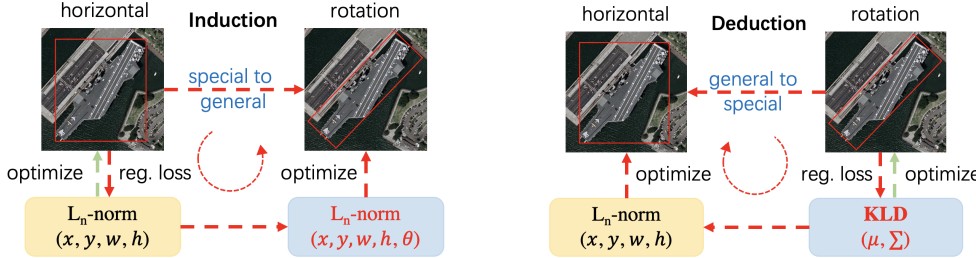

(a) Previous methods follow the induction paradigm from special horizontal to general rotated detection.

(b) Our proposed method adopts a deduction methodology from general rotated to special horizontal detection.

Figure 1: Methodological road-map difference between horizontal detection (special case) and rotation detection (general case) in the previous methods [1, 12, 13, 14, 15] and the proposed method.

In this paper, we take a step back, and aim to develop (from a deductive perspective) a unified regression framework for rotation detection and its special case: horizontal detection. In fact, our new framework enjoys a coherent property that it can be degenerated into the current commonly used regression loss (e.g. $l_n$-norm) in special cases (horizontal detection), as shown in Figure 1(b).

For a devising a rotation regression loss for high-precision rotation detection, one important observation is that the importance of different parameters to different types of objects can vary. For example, the angle parameter ($\theta$) and the center point parameter ($x, y$) are important for large aspect ratio objects and small objects, respectively. In another word, it is conjectured that regression loss should be self-modulated during the learning process and calls for more dynamic optimization strategy.

Inspired by the above ideas, we first convert the rotated bounding box $\mathcal{B}(x, y, h, w, \theta)$ into a 2-D Gaussian distribution $\mathcal{N}(\boldsymbol{\mu}, \boldsymbol{\Sigma})$. As a standard distance metric, we then use the Kullback-Leibler Divergence (KLD) [16] to calculate the distribution distance between the predicted bounding box and ground truth as the regression loss. We compare KLD with Smooth L1 loss [7] and another distance metric, Gaussian Wasserstein Distance (GWD) [5, 17], and find that KLD has a more complete parameter optimization mechanism. In particular, by analyzing the gradient of the parameters during learning, we show that the optimization of one parameter will be affected by other parameters (as the gradient weight). It means that the model will adaptively adjust the optimization strategy given a specific configuration of an object for detection, as shown can lead to excellent performance in high-precision detection. In addition, KLD is proven scale invariant, which is an important property that Smooth L1 loss and GWD do not possess. As the horizontal bounding box is a special case of the rotated bounding box, we show that KLD can also be degenerated into the $l_n$-norm loss as commonly used in existing horizontal detection pipeline. **The highlights of this paper are four-folds:**

**1)** Differing from the dominant existing practices that build rotation detectors heavily upon the horizontal detectors, we develop new rotation detection loss from scratch and show that it is coherent with existing horizontal detection protocol in its degenerated case for horizontal detection.

**2)** To achieve a more principled measurement between the prediction and ground truth, instead of computing the difference for each physically-meaningful parameter related to the bounding box which are in different scales and units, we innovatively convert the regression loss of rotation detection into the KLD of two 2-D Gaussian distributions, leading to a clean and coherent regression loss.

**3)** Through the gradient analysis of each parameter in KLD, we further find that the self-modulated optimization mechanism of KLD greatly promotes the improvement of high-precision detection, which verify the advantage of our loss design. More importantly, we have theoretically shown (in appendix) that KLD is scale invariant for detection, which is crucial for the rotation cases.

**4)** Extensive experimental results on seven public datasets and two popular detectors show the effectiveness of our approach, which achieves new state-of-the-art performance for rotation detection. The source codes [18] are made public available.

## 2   Background

We first generally discuss the related works on both horizontal and rotated object detection. Then we summarize the current design paradigm of rotation regression loss from two kinds of methodologies,

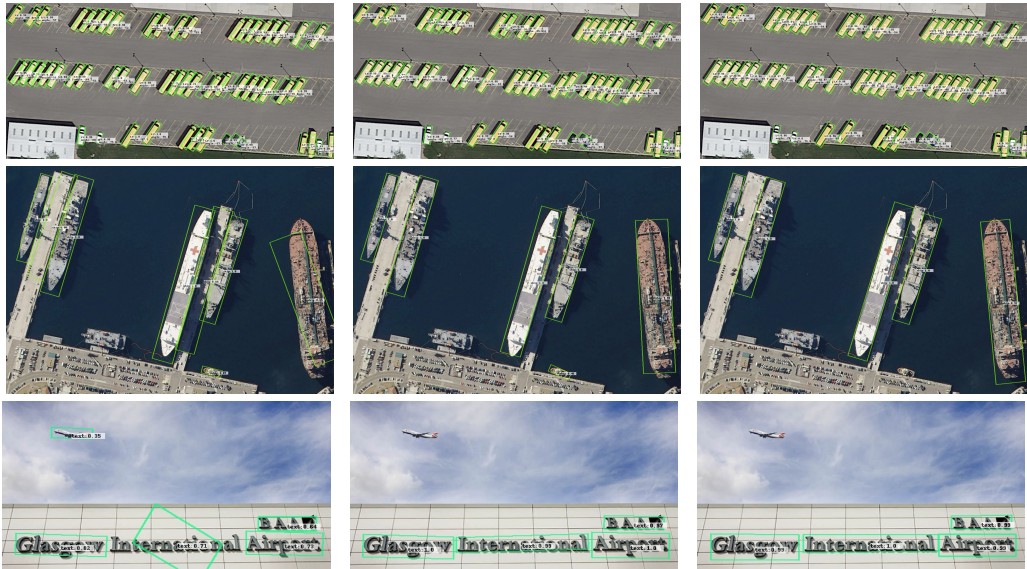

Figure 2: Visual comparison between Smooth L1 loss (left), GWD (middle) and KLD (right).

as shown in Figure 1: one is inductive that tries to develop the general rotation detection from the special and classic horizontal detection pipeline. While the other is deductive that aims to devise a general rotation detection pipeline with horizontal detection as its special case.

## 2.1 Related Works

**Horizontal object detection.** Horizontal object detection which covers most existing detection literature, normally uses a horizontal bounding box to represent the object. The mainstream classical object detection algorithms can be roughly divided according to the following standards: Two-[7, 8, 9, 11] or Single-stage [10, 19, 20] object detection, Anchor-free [21, 22, 23] or Anchor-based [8, 9, 10] object detection and CNN [8, 10, 21] or Transformer-based [24, 25] object detection. Although the pipelines may vary, the mainstream regression loss often uses the popular $l_n$-norm loss (such as smooth L1 loss) or IoU-based loss (such as GIoU [26], and DIoU [27]). These above-mentioned detectors have also been widely used in other scenarios and have achieved satisfactory performance. However, horizontal detectors do not provide accurate orientation and scale information.

**Rotated object detection.** Recent advances in rotation detection [3, 4, 12, 14, 28] are mainly driven by adapting the horizontal object detectors with rotated bounding boxes to represent multi-oriented objects. To accurately predict the rotated bounding box, most rotation detection methods extend the $l_n$-norm [12, 15, 29, 30, 31] used in horizontal detection, or construct a differentiable approximate IoU loss [3, 5, 32]. From scratch, we try to change the design of rotation regression loss from induction paradigm to deduction methodology, which in fact is a generalization to the horizontal case.

In the following, we describe the existing works from the induction and deduction methodologies.

## 2.2 Inductive Thinking of Loss Design: from Special Horizon to General Rotation Detection

Regression loss is a vital part of most current object detection algorithms. For horizontal bounding box regression, the model [7, 8, 9, 10, 11] mainly outputs four items for location and size:

$$t_x^p = \frac{x_p - x_a}{w_a}, t_y^p = \frac{y_p - y_a}{h_a}, t_w^p = \ln\left(\frac{w_p}{w_a}\right), t_h^p = \ln\left(\frac{h_p}{h_a}\right) \tag{1}$$

to match the four targets from the ground truth

$$t_x^t = \frac{x_t - x_a}{w_a}, t_y^t = \frac{y_t - y_a}{h_a}, t_w^t = \ln\left(\frac{w_t}{w_a}\right), t_h^t = \ln\left(\frac{h_t}{h_a}\right) \tag{2}$$

where $x, y, h, w$ denote the center coordinates, height and width, respectively. Variables $x_t, x_a, x_p$ are for the ground-truth box, anchor box, and predicted box, respectively (likewise for $y, w, h$).

Extending the above horizontal case, existing rotation detection models [1, 12, 13, 14, 15] also use regression loss which simply involves an extra angle parameter $\theta$:

$$t_\theta^p = f(\theta_p - \theta_a), t_\theta^t = f(\theta_t - \theta_a) \tag{3}$$

where $f(\cdot)$ is used to deal with angular periodicity, such as trigonometric functions, modulo, etc.

The overall regression loss for rotation detection is:

$$L_{reg} = l_n\text{-norm}\left(\Delta t_x, \Delta t_y, \Delta t_w, \Delta t_h, \Delta t_\theta\right) \tag{4}$$

where $\Delta t_x = t_x^p - t_x^t = \frac{\Delta x}{w_a}, \Delta t_y = t_y^p - t_y^t = \frac{\Delta y}{h_a}, \Delta t_w = t_w^p - t_w^t = \ln(w_p/w_t), \Delta t_h = t_h^p - t_h^t = \ln(h_p/h_t)$, and $\Delta t_\theta = t_\theta^p - t_\theta^t = \Delta\theta$.

It can be seen that parameters are optimized independently, making the loss (or detection accuracy) sensitive to the under-fitting of any of the parameters. This mechanism is fatal to high-precision detection. Taking the left side of Figure 2 as an example, the detection result based on the Smooth L1 loss often shows the deviation of the center point or angle. Moreover, different types of objects have different sensitivity to these five parameters. For example, the angle parameter is very important for detecting objects with large aspect ratios. This requires to select an appropriate set of weights given a specific single object sample during the training, which is nontrivial or even unrealistic.

## 2.3 Deductive Thinking of Loss Design: from General Rotation to Special Horizon Detection

To break the original inductive design paradigm, we adopt deductive paradigm to construct more accurate rotation regression loss. Here we rephrase the main idea in the recent work [5], which converts a arbitrary-oriented bounding box $\mathcal{B}(x, y, h, w, \theta)$ into a 2-D Gaussian $\mathcal{N}(\boldsymbol{\mu}, \boldsymbol{\Sigma})$, as illustrated in Figure 3. Then the distance between two Gaussian is calculated as the final loss. Specifically, the conversion is:

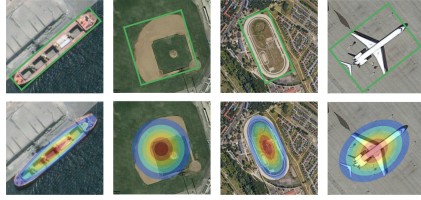

Figure 3: **Top:** rotated box $\mathcal{B}(x, y, h, w, \theta)$. **Bottom:** 2-D Gaussian dist. $\mathcal{N}(\boldsymbol{\mu}, \boldsymbol{\Sigma})$.

$$\boldsymbol{\mu} = (x, y)^\top$$

$$\boldsymbol{\Sigma}^{1/2} = \mathbf{R}\boldsymbol{\Lambda}\mathbf{R}^\top = \left(\begin{array}{cc} \cos\theta & -\sin\theta \\ \sin\theta & \cos\theta \end{array}\right)\left(\begin{array}{cc} \frac{w}{2} & 0 \\ 0 & \frac{h}{2} \end{array}\right)\left(\begin{array}{cc} \cos\theta & \sin\theta \\ -\sin\theta & \cos\theta \end{array}\right)$$

$$= \left(\begin{array}{cc} \frac{w}{2}\cos^2\theta + \frac{h}{2}\sin^2\theta & \frac{w-h}{2}\cos\theta\sin\theta \\ \frac{w-h}{2}\cos\theta\sin\theta & \frac{w}{2}\sin^2\theta + \frac{h}{2}\cos^2\theta \end{array}\right) \tag{5}$$

where $\mathbf{R}$ represents the rotation matrix, and $\boldsymbol{\Lambda}$ represents the diagonal matrix of eigenvalues.

The recent work [5] analyzes that the introduction of $\mathcal{N}(\boldsymbol{\mu}, \boldsymbol{\Sigma})$ can solve the inconsistency between metric and loss, boundary discontinuity and square-like problem. On this basis, we further study how to design high-precision detection regression loss through new parameter space. Our view is that the self-modulated mechanism is positively correlated with the final high-precision performance.

**Gaussian Wasserstein Distance.** The Wasserstein distance [5, 17] between two probability measures $\mathbf{X}_p \sim \mathcal{N}_p(\boldsymbol{\mu}_p, \boldsymbol{\Sigma}_p)$ and $\mathbf{X}_t \sim \mathcal{N}_t(\boldsymbol{\mu}_t, \boldsymbol{\Sigma}_t)$ expressed as:

$$\mathbf{D}_w(\mathcal{N}_p, \mathcal{N}_t)^2 = \underbrace{\|\boldsymbol{\mu}_p - \boldsymbol{\mu}_t\|_2^2}_{\text{center distance}} + \underbrace{\mathbf{Tr}(\boldsymbol{\Sigma}_p + \boldsymbol{\Sigma}_t - 2(\boldsymbol{\Sigma}_p^{1/2}\boldsymbol{\Sigma}_t\boldsymbol{\Sigma}_p^{1/2})^{1/2})}_{\text{coupling terms about } h_p,\, w_p \text{ and } \theta_p} \tag{6}$$

Eq. 6 shows that the Gaussian Wasserstein Distance (GWD) is mainly divided into two parts: the distance between the center points $(x, y)$ and the coupling terms about $h$, $w$ and $\theta$. Accordingly, the regression loss based on GWD can be regarded as a semi-coupled loss. Although GWD can greatly improve the performance of high-precision rotation detection due to the coupling between part of the parameters, the independent optimization of the center point make the detection result slightly shifted (see Figure 2). Note that GWD is not scale invariant, which is not detection friendly.

When all the boxes are horizontal ($\theta = 0°$), Eq. 6 can be further simplified:

$$\begin{aligned} \mathbf{D}_w^h(\mathcal{N}_p, \mathcal{N}_t)^2 &= \|\boldsymbol{\mu}_p - \boldsymbol{\mu}_t\|_2^2 + \|\boldsymbol{\Sigma}_p^{1/2} - \boldsymbol{\Sigma}_t^{1/2}\|_F^2 \\ &= (x_p - x_t)^2 + (y_p - y_t)^2 + \left((w_p - w_t)^2 + (h_p - h_t)^2\right)/4 \\ &= l_2\text{-norm}(\Delta x, \Delta y, \Delta w/2, \Delta h/2) \end{aligned} \tag{7}$$

where $\| \cdot \|_F$ is the Frobenius norm. Although Eq. 7 can still be used as the regression loss of horizontal detection, Eq. 4 and 7 are not completely consistent.

Although GWD scheme has played a preliminary exploration of the deductive paradigm, it does not focus on achieving high-precision detection and scale invariance. In the following, we will propose our new approach based on the Kullback-Leibler divergence (KLD) [16].

## 3 Proposed Approach

**Kullback-Leibler Divergence.** To explore the more appropriate regression loss, we adopt the Kullback-Leibler divergence (KLD) [16]. Similarly, the KLD between two 2-D Gaussian is:

$$\mathbf{D}_{kl}(\mathcal{N}_p||\mathcal{N}_t) = \underbrace{\frac{1}{2}(\boldsymbol{\mu}_p - \boldsymbol{\mu}_t)^\top \boldsymbol{\Sigma}_t^{-1}(\boldsymbol{\mu}_p - \boldsymbol{\mu}_t)}_{\text{term about } x_p \text{ and } y_p} + \underbrace{\frac{1}{2}\mathbf{Tr}(\boldsymbol{\Sigma}_t^{-1}\boldsymbol{\Sigma}_p) + \frac{1}{2}\ln\frac{|\boldsymbol{\Sigma}_t|}{|\boldsymbol{\Sigma}_p|}}_{\text{coupling terms about } h_p, w_p \text{ and } \theta_p} - 1 \tag{8}$$

or

$$\mathbf{D}_{kl}(\mathcal{N}_t||\mathcal{N}_p) = \underbrace{\frac{1}{2}(\boldsymbol{\mu}_p - \boldsymbol{\mu}_t)^\top \boldsymbol{\Sigma}_p^{-1}(\boldsymbol{\mu}_p - \boldsymbol{\mu}_t) + \frac{1}{2}\mathbf{Tr}(\boldsymbol{\Sigma}_p^{-1}\boldsymbol{\Sigma}_t) + \frac{1}{2}\ln\frac{|\boldsymbol{\Sigma}_p|}{|\boldsymbol{\Sigma}_t|}}_{\text{chain coupling of all parameters}} - 1 \tag{9}$$

It can be seen that each item in $\mathbf{D}_{kl}(\mathcal{N}_t||\mathcal{N}_p)$ is composed of partial parameter coupling, which makes all parameters form a chain coupling relationship. In the optimization process of the KLD-based detector, the parameters influence each other and are jointly optimized which make optimization mechanism of the model is self-modulated. In contrast, $\mathbf{D}_{kl}(\mathcal{N}_p||\mathcal{N}_t)$ and GWD are both semi-coupled, but $\mathbf{D}_{kl}(\mathcal{N}_p||\mathcal{N}_t)$ has a better central point optimization mechanism.

Although KLD is asymmetric, we find that the optimization principles of these two forms are similar by analyzing the gradients of various parameters and experimental results. Take the relatively simple $\mathbf{D}_{kl}(\mathcal{N}_p||\mathcal{N}_t)$ as an example, according to Eq. 5, each item of Eq. 8 can be expressed as

$$(\boldsymbol{\mu}_p - \boldsymbol{\mu}_t)^\top \boldsymbol{\Sigma}_t^{-1}(\boldsymbol{\mu}_p - \boldsymbol{\mu}_t) = \frac{4\left(\Delta x \cos\theta_t + \Delta y \sin\theta_t\right)^2}{w_t^2} + \frac{4\left(\Delta y \cos\theta_t - \Delta x \sin\theta_t\right)^2}{h_t^2} \tag{10}$$

$$\mathbf{Tr}(\boldsymbol{\Sigma}_t^{-1}\boldsymbol{\Sigma}_p) = \frac{h_p^2}{w_t^2}\sin^2\Delta\theta + \frac{w_p^2}{h_t^2}\sin^2\Delta\theta + \frac{h_p^2}{h_t^2}\cos^2\Delta\theta + \frac{w_p^2}{w_t^2}\cos^2\Delta\theta \tag{11}$$

$$\ln\frac{|\boldsymbol{\Sigma}_t|}{|\boldsymbol{\Sigma}_p|} = \ln\frac{h_t^2}{h_p^2} + \ln\frac{w_t^2}{w_p^2} \tag{12}$$

where $\Delta x = x_p - x_t, \Delta y = y_p - y_t, \Delta\theta = \theta_p - \theta_t$.

**Analysis of high-precision detection.** Without loss of generality, we set $\theta_t = 0°$, then

$$\frac{\partial \mathbf{D}_{kl}(\mu_p)}{\partial \mu_p} = \left(\frac{4}{w_t^2}\Delta x, \frac{4}{h_t^2}\Delta y\right)^\top \tag{13}$$

The weights $1/w_t^2$ and $1/h_t^2$ will make the model dynamically adjust the optimization of the object position according to the scale. For example, when the object scale is small or an edge is too short, the model will pay more attention to the optimization of the offset of the corresponding direction. For this kind of object, a slight deviation on the corresponding direction will often cause a sharp drop in IoU. When $\theta_t \neq 0°$, the gradient of the object offset ($\Delta x$ and $\Delta y$) will be dynamically adjusted according to the $\theta_t$ for better optimization. In contrast, the gradient of the center point in GWD and $L_2$-norm are $\frac{\partial \mathbf{D}_w(\mu_p)}{\partial \mu_p} = (2\Delta x, 2\Delta y)^\top$ and $\frac{\partial L_2(\mu_p)}{\partial \mu_p} = (\frac{2}{w_a^2}\Delta x, \frac{2}{h_a^2}\Delta y)^\top$. The former cannot adjust the dynamic gradient according to the length and width of the object. The latter is based on the length and width of the anchor ($w_a, h_a$) to adjust the gradient instead of the target object ($w_t, h_t$), which is almost ineffective for those detectors [3, 13, 15, 28, 29, 33, 34] that use horizontal anchors for rotation detection. More importantly, they are not related to the angle of the target object. Therefore, the detection result of the GWD-based and $L_n$-norm models will show a slight deviation, while the detection result of the KLD-based model is quite accurate, as shown in Figure 2.

For $h_p$ and $w_p$, we have

$$\frac{\partial \mathbf{D}_{kl}(\boldsymbol{\Sigma}_p)}{\partial \ln h_p} = \frac{h_p^2}{h_t^2}\cos^2\Delta\theta + \frac{h_p^2}{w_t^2}\sin^2\Delta\theta - 1, \quad \frac{\partial \mathbf{D}_{kl}(\boldsymbol{\Sigma}_p)}{\partial \ln w_p} = \frac{w_p^2}{w_t^2}\cos^2\Delta\theta + \frac{w_p^2}{h_t^2}\sin^2\Delta\theta - 1 \tag{14}$$

On the one hand, the optimization of the $h_p$ and $w_p$ is affected by the $\Delta\theta$. When $\Delta\theta = 0°$, $\frac{\partial \mathbf{D}_{kl}(\mathbf{\Sigma}_p)}{\partial \ln h_p} = \frac{h_p^2}{h_t^2} - 1$, $\frac{\partial \mathbf{D}_{kl}(\mathbf{\Sigma}_p)}{\partial \ln w_p} = \frac{w_p^2}{w_t^2} - 1$, which means that the smaller targeted height or width leads to heavier penalty on its matching loss. This is desirable, as smaller height or width needs higher matching precision. On the other hand, the optimization of $\Delta\theta$ is also affected by $h_p$ and $w_p$:

$$\frac{\partial \mathbf{D}_{kl}(\mathbf{\Sigma}_p)}{\partial \theta_p} = \left( \frac{h_p^2 - w_p^2}{w_t^2} + \frac{w_p^2 - h_p^2}{h_t^2} \right) \sin 2\Delta\theta \tag{15}$$

when $w_p = w_t, h_p = h_t$, then $\frac{\partial \mathbf{D}_{kl}(\mathbf{\Sigma}_p)}{\partial \theta_p} = \left( \frac{h_t^2}{w_t^2} + \frac{w_t^2}{h_t^2} - 2 \right) \sin 2\Delta\theta \geq \sin 2\Delta\theta$, the condition for the equality sign is $h_t = w_t$. This shows that the larger the aspect ratio of the object, the model will pay more attention to the optimization of the angle. This is the main reason why the KLD-based model has a huge advantage in high-precision detection indicators as a slight angle error would cause a serious accuracy drop for large aspect ratios objects. Through the above analysis, we find that when one of the parameters is optimized, the other parameters will be used as its weight to dynamically adjust the optimization rate. In other words, the optimization of parameters is no longer independent, that is, optimizing one parameter will also promote the optimization of other parameters. The optimization of this virtuous circle is the key to KLD as an excellent rotation regression loss. In addition, $\mathbf{D}_{kl}(\mathcal{N}_t||\mathcal{N}_p)$ has similar properties, refer to appendix for details.

**Scale invariance.** For a full-rank matrix $\mathbf{M}$, $|\mathbf{M}| \neq 0$, we have $\mathbf{D}_{kl}(\mathcal{N}_p||\mathcal{N}_t) = \mathbf{D}_{kl}(\mathcal{N}_{p'}||\mathcal{N}_{t'})$, where $\mathbf{X}_{p'} = \mathbf{M}\mathbf{X}_p \sim \mathcal{N}_p(\mathbf{M}\boldsymbol{\mu}_p, \mathbf{M}\mathbf{\Sigma}_p\mathbf{M}^\top)$, $\mathbf{X}_{t'} = \mathbf{M}\mathbf{X}_t \sim \mathcal{N}_t(\mathbf{M}\boldsymbol{\mu}_t, \mathbf{M}\mathbf{\Sigma}_t\mathbf{M}^\top)$. Therefore, the affine invariance (including scale invariance when $\mathbf{M} = k\mathbf{I}$, where $\mathbf{I}$ denotes identity matrix) of KLD can be proven (see proof in appendix). Compared with $L_n$-norm and GWD, KLD is more suitable for replacing the non-differentiable rotated IoU loss for its consistency with detection metric.

**Horizontal special case.** For horizontal detection, combine Eq. 8 to Eq. 12, we have

$$\begin{aligned}
\mathbf{D}_{kl}^h(\mathcal{N}_p||\mathcal{N}_t) &= \frac{1}{2} \left( \frac{w_p^2}{w_t^2} + \frac{h_p^2}{h_t^2} + \frac{4\Delta^2 x}{w_t^2} + \frac{4\Delta^2 y}{h_t^2} + \ln \frac{w_t^2}{w_p^2} + \ln \frac{h_t^2}{h_p^2} - 2 \right) \\
&= 2l_2\text{-norm}(\Delta t_x, \Delta t_y) + l_1\text{-norm}(\Delta t_w, \Delta t_h) + \frac{1}{2} l_2\text{-norm}(\frac{1}{\Delta t_w}, \frac{1}{\Delta t_h}) - 1
\end{aligned} \tag{16}$$

where the first two terms of Eq. 16 are very similar to Eq. 4, and the divisor part of the two terms $x$ and $y$ is the main difference ($\frac{\Delta x}{w_t}$ vs. $\frac{\Delta x}{w_a}$).

**Variants of KLD.** We have also introduced some variants [35, 36] of KLD to further verify the influence of asymmetry on rotation detection can be ignored. The variants mainly including

$$\begin{aligned}
\mathbf{D}_{kl\_min(max)}(\mathcal{N}_p||\mathcal{N}_t) &= \min(\max)\left(\mathbf{D}_{kl}(\mathcal{N}_p||\mathcal{N}_t), \mathbf{D}_{kl}(\mathcal{N}_t||\mathcal{N}_p)\right) \\
\mathbf{D}_{js}(\mathcal{N}_p||\mathcal{N}_t) &= \frac{1}{2} \left( \mathbf{D}_{kl}\left( \mathcal{N}_t || \frac{\mathcal{N}_p + \mathcal{N}_t}{2} \right) + \mathbf{D}_{kl}\left( \mathcal{N}_p || \frac{\mathcal{N}_p + \mathcal{N}_t}{2} \right) \right) \\
\mathbf{D}_{jef}(\mathcal{N}_p||\mathcal{N}_t) &= \mathbf{D}_{kl}(\mathcal{N}_t||\mathcal{N}_p) + \mathbf{D}_{kl}(\mathcal{N}_p||\mathcal{N}_t)
\end{aligned} \tag{17}$$

**Rotation regression loss.** The whole training process of detector is as follows: i) predict offset $(t_x^p, t_y^p, t_w^p, t_h^p, t_\theta^p)$; ii) decode prediction box; iii) convert prediction box and target ground-truth into Gaussian distribution; iv) calculate KLD of two Gaussian distributions. Therefore, the inference time remains unchanged. We normalize the distance function as our final regression loss $\mathcal{L}_{reg}$:

$$\mathcal{L}_{reg} = 1 - \frac{1}{\tau + f(\mathbf{D})}, \quad \tau \geq 1 \tag{18}$$

where $f(\cdot)$ denotes a non-linear function to transform the distance $\mathbf{D}$ to make the loss more smooth and expressive. In this paper, we mainly use two nonlinear functions, $sqrt(\mathbf{D})$ and $\ln(\mathbf{D} + 1)$. The hyperparameter $\tau$ modulates the entire loss. The multi-task loss is:

$$\mathcal{L} = \frac{\lambda_1}{N_{pos}} \sum_{n=1}^{N_{pos}} \mathcal{L}_{reg}(b_n, gt_n) + \frac{\lambda_2}{N} \sum_{n=1}^{N} \mathcal{L}_{cls}(p_n, t_n) \tag{19}$$

where $N_{pos}$ and $N$ indicate the number of positive and all anchors. $b_n$ denotes the $n$-th bounding box, $gt_n$ is the $n$-th target ground-truth. $t_n$ denotes the label of $n$-th object, $p_n$ is the $n$-th probability distribution of various classes calculated by sigmoid function. The hyper-parameter $\lambda_1, \lambda_2$ control the trade-off and are set to $\{2, 1\}$ by default. The classification loss $L_{cls}$ is set as focal loss [10].

Table 1: Ablation study of the loss form and hyperparameter on HRSC2016.

| Loss | $\mathbf{D}_{kl}$ | $f(\mathbf{D}_{kl})$ | $\mathcal{L}_G(f(\mathbf{D}_{kl}),\tau)$ | | | |
|---|---|---|---|---|---|---|
| | | | $\tau=1$ | $\tau=2$ | $\tau=3$ | $\tau=5$ |
| $f(\mathbf{D}_{kl})=sqrt(\mathbf{D}_{kl})$ | 0.20 | 82.96 | 84.85 | 84.15 | 75.23 | 73.32 |
| $f(\mathbf{D}_{kl})=\log(\mathbf{D}_{kl}+1)$ | | 83.23 | **85.25** | 83.63 | 80.79 | 73.44 |

Table 2: Ablation of different KLD-based regression loss form. The based detector is RetinaNet.

| Dataset | $\mathbf{D}_{kl}(\mathcal{N}_p\|\mathcal{N}_t)$ | $\mathbf{D}_{kl}(\mathcal{N}_t\|\mathcal{N}_p)$ | $\mathbf{D}_{kl\_min}(\mathcal{N}_p\|\mathcal{N}_t)$ | $\mathbf{D}_{kl\_max}(\mathcal{N}_p\|\mathcal{N}_t)$ | $\mathbf{D}_{js}(\mathcal{N}_p\|\mathcal{N}_t)$ | $\mathbf{D}_{jeffreys}(\mathcal{N}_p\|\mathcal{N}_t)$ |
|---|---|---|---|---|---|---|
| DOTA-v1.0 | 70.17 | 70.64 | **70.71** | 70.55 | 69.67 | 70.56 |
| HRSC2016 | 82.83 | 83.82 | 83.60 | 82.70 | **84.06** | 83.66 |

Table 3: Ablation study of normalization. The based detector is RetinaNet.

| Loss | Norm by Eq. 18 | HRSC2016 | | | DOTA-v1.0 |
|---|---|---|---|---|---|
| | | Hmean$_{50}$ | Hmean$_{75}$ | Hmean$_{50:95}$ | AP$_{50}$ |
| Smooth L1 | w/ | 78.99 | 43.12 | 43.47 | 64.95 |
| | w/o | **84.80** | **48.42** | **47.76** | **65.73** |

# 4 Experiment

## 4.1 Datasets and Implementation Details

Our experiments are conducted over a variety of datasets, including three large-scale public datasets for aerial images i.e. DOTA [37], UCAS-AOD [38], HRSC2016 [39], as well as scene text dataset ICDAR2015 [40], MLT [41] and MSRA-TD500 [42].

DOTA is one of the largest dataset for oriented object detection in aerial images with three released versions: DOTA-v1.0, DOTA-v1.5 and DOTA-v2.0. DOTA-v1.0 contains 15 common categories, 2,806 images and 188,282 instances. The proportions of the training set, validation set, and testing set in DOTA-v1.0 are 1/2, 1/6, and 1/3, respectively. In contrast, DOTA-v1.5 uses the same images as DOTA-v1.0, but extremely small instances (less than 10 pixels) are also annotated. Moreover, a new category, containing 402,089 instances in total is added in this version. While DOTA-v2.0 contains 18 common categories, 11,268 images and 1,793,658 instances. Compared to DOTA-v1.5, it further includes the new categories. The 11,268 images in DOTA-v2.0 are split into training, validation, test-dev, and test-challenge sets. We divide the images into $600 \times 600$ subimages with an overlap of 150 pixels and scale it to $800 \times 800$, in line with the cropping protocol in literature [5, 28].

UCAS-AOD contains 1,510 aerial images of approximately $659 \times 1,280$ pixels, with two categories of 14,596 instances in total. In line with [31, 37], we randomly select 1,110 for training and 400 for testing. HRSC2016 contains images from two scenarios including ships on sea and ships close inshore. The training, validation and test set include 436, 181 and 444 images.

ICDAR2015, MLT and MSRA-TD500 are commonly used for oriented scene text detection and spotting. ICDAR2015 includes 1,000 training images and 500 testing images. ICDAR2017 MLT is a multi-lingual text dataset, which includes 7,200 training images, 1,800 validation images and 9,000 testing images. MSRA-TD500 dataset consists of 300 training images and 200 testing images.

We use Tensorflow [43] to implement the proposed methods on a server with Tesla V100 and 32G memory. The experiments are all initialized by ResNet50 [44] by default unless otherwise specified. Weight decay and momentum are set 0.0001 and 0.9, respectively. We employ MomentumOptimizer over 8 GPUs with a total of 8 images per minibatch (1 image per GPU).

All the used datasets are trained by 20 epochs in total, and the learning rate is reduced tenfold at 12 epochs and 16 epochs, respectively. The initial learning rate is set to 5e-4. The number of image iterations per epoch for DOTA-v1.0, DOTA-v1.5, DOTA-v2.0, UCAS-AOD, HRSC2016, ICDAR2015, MLT and MSRA-TD500 are 54k, 64k, 80k, 5k, 10k, 10k, 10k and 5k respectively, and doubled if data augmentation (include random rotation, flipping, and graying) or multi-scale training is used.

## 4.2 Ablation Study and Further Comparison

**Regression loss form and hyperparameter.** Table 1 compares three forms of KLD-based regression loss on HRSC2016, including $\mathbf{D}_{kl}$, $f(\mathbf{D}_{kl})$ and $\mathcal{L}_{reg}(f(\mathbf{D}_{kl}),\tau)$. Due to extreme sensitivity to large

Table 4: High-precision detection experiment under different regression loss. 'R', 'F' and 'G' indicate random rotation, flipping, and graying, respectively. The resolution of HRSC2016, MSRA-TD500 and ICDAR2015 are $500 \times 500$, $800 \times 1,000$ and $800 \times 1,000$, respectively.

| Method | Dataset | Data Aug. | Reg. Loss | Hmean$_{50}$/AP$_{50}$ | Hmean$_{60}$/AP$_{60}$ | Hmean$_{75}$/AP$_{75}$ | Hmean$_{85}$/AP$_{85}$ | Hmean$_{50:95}$/AP$_{50:95}$ |
|---|---|---|---|---|---|---|---|---|
| RetinaNet | | | Smooth L1 | 84.28 | 74.74 | 48.42 | 12.56 | 47.76 |
| | | | GWD | 85.56 (+1.28) | 84.04 (+9.30) | 60.31 (+11.89) | 17.14 (+4.58) | 52.89 (+5.13) |
| | HRSC2016 | R+F+G | KLD | **87.45 (+3.17)** | **86.72 (+11.98)** | **72.39 (+23.97)** | **27.68 (+15.12)** | **57.80 (+10.04)** |
| R$^3$Det | | | Smooth L1 | 88.52 | 79.01 | 43.42 | 4.58 | 46.18 |
| | | | GWD | 89.43 (+0.91) | 88.89 (+9.88) | 65.88 (+22.46) | 15.02 (+10.44) | 56.07 (+9.89) |
| | | | KLD | **89.97 (+1.45)** | **89.73 (+10.72)** | **77.38 (+33.96)** | **25.12 (+20.54)** | **61.40 (+15.22)** |
| RetinaNet | MSRA-TD500 | R+F | Smooth L1 | 70.98 | 62.42 | 36.73 | 12.56 | 37.89 |
| | | | GWD | 76.76 (+5.78) | 68.58 (+6.16) | 44.21 (+7.48) | 17.75 (+5.19) | 43.62 (+5.73) |
| | | | KLD | **76.96 (+5.98)** | **70.08 (+7.66)** | **46.95 (+10.22)** | **19.59 (+7.03)** | **45.24 (+7.35)** |
| | ICDAR2015 | F | Smooth L1 | 69.78 | 64.15 | 36.97 | 8.71 | 37.73 |
| | | | GWD | 74.29 (+4.51) | 68.34 (+4.19) | 43.39 (+6.42) | 10.50 (+1.79) | 41.68 (+3.95) |
| | | | KLD | **75.32 (+5.54)** | **69.94 (+5.79)** | **44.46 (+7.49)** | **10.70 (+1.99)** | **42.68 (+4.95)** |
| | | R+F | Smooth L1 | 74.83 | 69.46 | 42.02 | 11.59 | 41.98 |
| | | | GWD | 76.15 (+1.32) | 71.26 (+1.80) | 45.59 (+3.57) | **11.65 (+0.06)** | 43.58 (+1.60) |
| | | | KLD | **77.92 (+3.09)** | **72.77 (+3.31)** | 43.27 (+1.25) | 11.09 (-0.50) | **43.65 (+1.67)** |
| R$^3$Det | | F | Smooth L1 | 74.28 | 68.12 | 35.73 | 8.01 | 39.10 |
| | | | GWD | 75.59 (+1.31) | 68.36 (+0.24) | 40.24 (+4.51) | 9.15 (+1.14) | 40.80 (+1.70) |
| | | | KLD | **77.72 (+2.43)** | **71.99 (+3.87)** | **43.95 (+8.22)** | **10.43 (+2.42)** | **43.29 (+4.19)** |
| | | R+F | Smooth L1 | 75.53 | 69.69 | 37.69 | 9.03 | 40.56 |
| | | | GWD | 77.09 (+1.56) | 71.52 (+1.83) | 41.08 (+3.39) | 10.10 (+1.07) | 42.17 (+1.61) |
| | | | KLD | **79.63 (+4.63)** | **73.30 (+3.61)** | **43.51 (+5.82)** | **10.61 (+1.58)** | **43.61 (+3.05)** |

errors, the performance of $\mathbf{D}_{kl}$ is extremely poor, only **0.20%**. Through a simple nonlinear linear transformation, the performance can be increased to **82.96%** and **83.23%** corresponding to $sqrt$ and $log$. We further perform a detailed hyperparameter experiment on the loss $\mathcal{L}_{reg}$ proposed in this paper, and the performance reaches the optimal when $\tau = 1$, $f(\mathbf{D}_{kl}) = \log(\mathbf{D}_{kl} + 1)$, about **85.25%**. Keeping the same loss pattern, we compare six KLD-based distance functions in Table 2, and conclude that the asymmetry of KLD does not have much impact on performance. In subsequent experiments, we use $\mathcal{L}_{reg}(\log(\mathbf{D}_{kl}(\mathcal{N}_p||\mathcal{N}_t)), 1)$ as the basic setting.

**Ablation study of normalization.** As mentioned above, the use of Eq. 18 is to smooth its excessively rapid growth trend and play a role of normalization. This extra normalization questions if the KLD is actually contributing or simply produces noise in the results. In order to further prove that our method is indeed effective, we also perform a normalization operation on the Smooth L1 loss to eliminate the interference caused by normalization. As shown in Table 3, there is a significant drop in performance after using the normalization. The above experimental results prove that the effectiveness of KLD does not come from Eq. 18.

**High-precision detection experiment.** We expect that the designed rotation regression loss can show advantages in high-precision detection. Table 4 shows the comparison of the high-precision detection results of three different regression losses using Smooth L1, GWD and KLD on different datasets and different detectors. For the HRSC2016 dataset containing a large number of ship with large aspect ratios, GWD-based RetinaNet has a **11.89%** improvement over Smooth L1 on AP$_{75}$, KLD even gets a **23.97%** gain. Even with a stronger R$^3$Det detector, KLD and GWD still increased by **33.96%** and **22.46%** in AP$_{75}$, and **15.22%** and **9.89%** in AP$_{50:95}$. The same experimental conclusion are also reflected in the other two scene text datasets MASR-TF500 and ICDAR2015, which is KLD > GWD > Smooth L1. In general, the self-modulation optimization mechanism has a significant help for high-precision detection. For a more intuitive comparison, we visually compare these three regression losses, as shown in Figure 2. Since the center point $(x, y)$ parameters in Smooth L1 Loss and GWD are independently optimized, their prediction results are slightly shifted. In contrast, the KLD-based prediction results are closer to the object boundary and show strong robustness in dense scenes. Similarly, GWD-based or KLD-based model has more accurate angle prediction capabilities than Smooth L1-based model due to their angle parameters $(\theta)$ are not independently optimized.

**Ablation study on more datasets.** To make the results more credible, we continue to verify on the other five datasets, as shown in Table 5. The improvement of KLD on the three data sets of MLT, UCAS-AOD and DOTA-v1.0 is still considerable, with an increase of **9.17%**, **1.58%**, and **5.55%** respectively. Note that for DOTA-v1.5 and DOTA-v2.0, which contain a large number of small objects (less than 10 pixels), KLD has achieved significant gains of **3.63%** and **3.53%**.

**Comparison of peer methods.** Table 6 compares the six peer techniques, including IoU-Smooth L1 Loss [3], Modulated loss [45], RIL [34], CSL [4, 47], DCL [46], and GWD [5] on DOTA-v1.0. For fairness, these methods are all implemented on the same baseline method, and are trained and tested under the same environment and hyperparameters. We detail the accuracy of the seven categories,

Table 5: More ablation experiments on other datasets.

| Method | Reg. Loss | MLT | UCAS-AOD | DOTA-v1.0 | DOTA-v1.5 | DOTA-v2.0 |
|---|---|---|---|---|---|---|
| | Smooth L1 | 48.42 | 94.56 | 65.73 | 58.87 | 44.16 |
| RetinaNet | GWD | 54.58 (+6.16) | 95.44 (+0.88) | 68.93 (+3.20) | 60.03 (+1.16) | 46.65 (+2.49) |
| | KLD | **57.59** (+9.17) | **96.14** (+1.58) | **71.28** (+5.55) | **62.50** (+3.63) | **47.69** (+3.53) |

Table 6: Accuracy comparison between different rotation detectors on DOTA dataset. † and ‡ represent the large aspect ratio object and the square-like object, respectively. The bold **red** and **blue** fonts indicate the top two performances respectively. $D_{oc}$ and $D_{le}$ represent OpenCV Definition ($\theta \in [-90°, 0°)$) and Long Edge Definition ($\theta \in [-90°, 90°)$) of RBox.

| Baseline | Method | Box Def. | v1.0 tranval/test | | | | | | | | | v1.0 train/val | | | v1.5 | v2.0 |
|---|---|---|---|---|---|---|---|---|---|---|---|---|---|---|---|---|
| | | | BR† | SV† | LV† | SH† | HA† | ST‡ | RA‡ | 7-AP50 | AP50 | AP50 | AP75 | AP50:95 | AP50 | AP50 |
| RetinaNet | - | $D_{oc}$ | 42.17 | 65.93 | 51.11 | 72.61 | 53.24 | 78.38 | 62.00 | 60.78 | 65.73 | 64.70 | 32.31 | 34.50 | 58.87 | 44.16 |
| | - | $D_{le}$ | 38.31 | 60.48 | 49.77 | 68.29 | 51.28 | 78.60 | 60.02 | 58.11 | 64.17 | 62.21 | 26.06 | 31.49 | 56.10 | 43.06 |
| | IoU-Smooth L1 [3] | $D_{oc}$ | 44.32 | 63.03 | 51.25 | 72.78 | 56.21 | 77.98 | 63.22 | 61.26 | 66.99 | 64.61 | 34.17 | 36.23 | 59.16 | 46.31 |
| | Modulated Loss [45] | $D_{oc}$ | 42.92 | 67.92 | 52.91 | 72.67 | 53.64 | **80.22** | 58.21 | 61.21 | 66.05 | 63.50 | 33.32 | 34.61 | 57.75 | 45.17 |
| | Modulated Loss [45] | Quad. | 43.21 | 70.78 | 54.70 | 72.68 | **60.99** | 79.72 | 62.08 | 63.45 | 67.20 | 65.15 | 40.59 | **39.12** | **61.42** | **46.71** |
| | RIL [34] | Quad. | 40.81 | 67.63 | 55.45 | 72.42 | 55.49 | 78.09 | **64.75** | 62.09 | 66.06 | 64.07 | **40.98** | 39.05 | 58.91 | 45.35 |
| | CSL [4] | $D_{le}$ | 42.25 | 68.28 | 54.51 | 72.85 | 53.10 | 75.59 | 58.99 | 60.80 | 67.38 | 64.40 | 32.58 | 35.04 | 58.55 | 43.34 |
| | DCL (BCL) [46] | $D_{le}$ | 41.40 | 65.82 | 56.27 | 73.80 | 54.30 | 79.02 | 60.25 | 61.55 | 67.39 | **65.93** | 35.66 | 36.71 | 59.38 | 45.46 |
| | GWD [5] | $D_{oc}$ | **44.07** | 71.92 | 62.56 | 77.94 | 60.25 | 79.64 | 63.52 | **65.70** | **68.93** | 68.14 | 38.68 | 38.71 | 60.03 | 46.65 |
| | KLD | $D_{oc}$ | **44.00** | **74.45** | **72.48** | **84.30** | 65.54 | 80.03 | 65.05 | **69.41** | **71.28** | 68.14 | **44.48** | **42.15** | **62.50** | **47.69** |
| R³Det | - | $D_{oc}$ | 44.15 | 75.09 | 72.88 | 86.04 | 56.49 | 82.53 | 61.01 | 68.31 | 70.66 | 67.18 | 38.41 | 38.46 | 62.91 | 48.43 |
| | DCL (BCL) [46] | $D_{le}$ | **46.84** | 74.87 | 74.96 | 85.70 | 57.72 | **84.06** | **63.77** | 69.70 | 71.21 | 67.45 | 35.44 | 37.54 | 61.98 | 48.71 |
| | GWD [5] | $D_{oc}$ | 46.73 | **75.84** | **78.00** | **86.71** | 62.69 | **83.09** | 61.12 | **70.60** | **71.56** | **69.28** | **43.35** | **41.56** | **63.22** | **49.25** |
| | KLD | $D_{oc}$ | **48.34** | 75.09 | **78.88** | 86.52 | 65.48 | 82.08 | 61.51 | **71.13** | **71.73** | 68.87 | **44.48** | 42.11 | **65.18** | **50.90** |

including large aspect ratio (e.g. BR, SV, LV, SH, HA) and square-like object (e.g. ST, RD), which can better reflect the real-world challenges and advantages of our method. Without bells and whistles, the combination of RetinaNet and KLD directly surpasses R³Det (**71.28%** vs. **70.66%** in $AP_{50}$ and **69.41%** vs. **68.31%** in 7-$AP_{50}$). Even combined with R³Det, KLD can still further improve performance of the large aspect ratio object (**2.82%** in 7-$AP_{50}$) and high-precision detection (**6.07%** in $AP_{75}$ and **3.65%** $AP_{50:95}$). KLD-based method shows the best performer in almost all indicators. Similar conclusions can still be drawn on the more challenging datasets (DOTA-v1.5 and DOTA-v2.0), which contain more data and tiny object (less than 10 pixels).

**Horizontal detection verification.** As analyzed by Eq. 16, KLD can be degenerated into the common regression loss in horizontal detection task. Table 7 compares the regression loss Smooth L1 and IoU/GIoU for horizontal detection with the proposed regression loss KLD on MS COCO [48] dataset. The results show that our KLD is not worse than other losses on the Faster RCNN [8], RetinaNet [10] and FCOS [21], and even has an improvement of **0.6%** on RetinaNet. The ground truth for rotation detection is the minimum circumscribed rectangle, which means that ground truth can well reflect the true scale and direction information of the object. The "horizontal special case" described in this paper also meets the above requirements, the horizontal circumscribed rectangle is equal to the minimum circumscribed rectangle at this time. Although the ground truth of the COCO is a horizontal box, it is not the minimum circumscribed rectangle, which means that it loses the direction information and accurate scale information of the object. For example, a baseball bat placed obliquely in the image, the height and width of its horizontal circumscribed rectangle do not represent the height and width of the object itself. This causes that when KLD is applied to the COCO, the optimization mechanism of KLD that dynamically adjusts the angle gradient according to the aspect ratio is meaningless, which affects the improvement of the final performance. In general, this is a defect in the dataset annotation itself, not that KLD is not good enough. In fact, it is inappropriate to use the COCO to discuss $\theta = 0°$, because the COCO discards $\theta$ parameter. In addition, $\theta = 0°$ describes the instances in the horizontal position, but not mean all instances of the dataset are in a horizontal position. This paper uses COCO to discuss the "horizontal special case" to express that even if the dataset has certain labeling defects, KLD can have certain effects. After all, it is difficult to observe the performance improvement of all horizontal objects on the rotating dataset.

## 4.3 Comparisons with the State-of-the-Art Methods

The evaluation is performed on the DOTA, which contains a considerable number of categories, complexity scenes. Our single-scale model RetinaNet-KLD-R50 and R³Det-KLD-R50 achieve **75.28%** and **77.36%** respectively. They outperform multi-scale models as shown in Table 8. With large backbone and multi-scale testing, our method further achieves state-of-the-art accuracy **80.63%**.

Table 7: Performance evaluation of KLD on classic horizontal detection.

| Detector | Regression Loss | AP | AP$_{50}$ | AP$_{75}$ | AP$_s$ | AP$_m$ | AP$_l$ |
|---|---|---|---|---|---|---|---|
| RetinaNet [10] | Smooth L1 | 37.2 | 56.6 | 39.7 | 21.4 | 41.1 | 48.0 |
| | GIoU | 37.4 | **56.7** | 39.7 | 22.2 | 41.7 | 48.1 |
| | KLD | **38.0** | 56.4 | **40.6** | **23.3** | **43.2** | **49.3** |
| Faster RCNN [8] | Smooth L1 | 37.9 | **58.8** | 41.0 | 22.4 | 41.4 | 49.1 |
| | GIoU | **38.3** | 58.7 | 41.5 | 22.5 | 41.7 | **49.7** |
| | KLD | 38.2 | 58.7 | **41.7** | **22.6** | **41.8** | 49.3 |
| FCOS [21] | IoU | 36.6 | 56.0 | 38.8 | 21.0 | 40.6 | 47.0 |
| | KLD | **36.8** | **56.3** | **39.1** | **21.7** | **40.8** | **47.5** |

Table 8: AP on different objects on DOTA-v1.0. Here R-101 denotes ResNet-101 (likewise for R-50, R-152), and RX-101 and H-104 represent ResNeXt101 [49] and Hourglass-104 [50], respectively. MS indicates that multi-scale training/testing is used. **Red** and **blue** indicate the top two performances.

| | Method | Backbone | MS | PL | BD | BR | GTF | SV | LV | SH | TC | BC | ST | SBF | RA | HA | SP | HC | AP$_{50}$ |
|---|---|---|---|---|---|---|---|---|---|---|---|---|---|---|---|---|---|---|---|
| Two-stage | ICN [31] | R-101 | ✓ | 81.40 | 74.30 | 47.70 | 70.30 | 64.90 | 67.80 | 70.00 | 90.80 | 79.10 | 78.20 | 53.60 | 62.90 | 67.00 | 64.20 | 50.20 | 68.20 |
| | RoI-Trans. [12] | R-101 | ✓ | 88.64 | 78.52 | 43.44 | 75.92 | 68.81 | 73.68 | 83.59 | 90.74 | 77.27 | 81.46 | 58.39 | 53.54 | 62.83 | 58.93 | 47.67 | 69.56 |
| | SCRDet [3] | R-101 | ✓ | 89.98 | 80.65 | 52.09 | 68.36 | 68.36 | 60.32 | 72.41 | 90.85 | **87.94** | 86.86 | 65.02 | 66.68 | 66.25 | 68.24 | 65.21 | 72.61 |
| | Gliding Vertex [51] | R-101 | | 89.64 | 85.00 | 52.26 | 77.34 | 73.01 | 73.14 | 86.82 | 90.74 | 79.02 | 86.81 | 59.55 | **70.91** | 72.94 | 70.86 | 57.32 | 75.02 |
| | Mask OBB [52] | RX-101 | ✓ | 89.56 | 85.95 | 54.21 | 72.90 | 76.52 | 74.16 | 85.63 | 89.85 | 83.81 | 86.48 | 54.89 | 69.64 | 73.94 | 69.06 | 63.32 | 75.33 |
| | CenterMap OBB [53] | R-101 | ✓ | 89.83 | 84.41 | 54.60 | 70.25 | 77.66 | 78.32 | 87.19 | 90.66 | 84.89 | 85.27 | 56.46 | 69.23 | 74.13 | 71.56 | 66.06 | 76.03 |
| | FPN-CSL [4] | R-152 | ✓ | **90.25** | **85.53** | 54.64 | 75.31 | 70.44 | 73.51 | 77.62 | 90.84 | 86.15 | 86.69 | 69.60 | 68.04 | 73.83 | 71.10 | 68.93 | 76.17 |
| | RSDet-II [45] | R-152 | ✓ | 89.93 | 84.45 | 53.77 | 74.35 | 71.52 | 78.31 | 78.12 | **91.14** | 87.35 | 86.93 | 65.64 | 65.17 | 75.35 | **79.74** | 63.31 | 76.34 |
| | SCRDet++ [54] | R-101 | ✓ | **90.05** | 84.39 | 55.44 | 73.99 | 77.54 | 71.11 | 86.05 | 90.67 | 87.32 | 87.08 | 69.62 | 68.90 | 73.74 | 71.29 | 65.08 | 76.81 |
| | ReDet [55] | ReR-50 | ✓ | 88.81 | 82.48 | **60.83** | 80.82 | 78.34 | **86.06** | 88.31 | 90.87 | **88.77** | 87.03 | 68.65 | 66.90 | **79.26** | **79.71** | 74.67 | 80.10 |
| Single-stage | PIoU [32] | DLA-34 [56] | | 80.90 | 69.70 | 24.10 | 60.20 | 38.30 | 64.40 | 64.80 | **90.90** | 77.20 | 70.40 | 46.50 | 37.10 | 57.10 | 61.90 | 64.00 | 60.50 |
| | O$^2$-DNet [57] | H-104 | ✓ | 89.31 | 82.14 | 47.33 | 61.21 | 71.32 | 74.03 | 78.62 | 90.76 | 82.23 | 81.36 | 60.93 | 60.17 | 58.21 | 66.98 | 61.03 | 71.04 |
| | DAL [15] | R-101 | ✓ | 88.61 | 79.69 | 46.27 | 70.37 | 65.89 | 76.10 | 78.53 | 90.84 | 79.98 | 78.41 | 58.71 | 62.02 | 69.23 | 71.32 | 60.65 | 71.78 |
| | P-RSDet [58] | R-101 | ✓ | 88.58 | 77.83 | 50.44 | 69.29 | 71.10 | 75.79 | 78.66 | 90.88 | 80.10 | 81.71 | 57.92 | 63.03 | 66.30 | 69.77 | 63.13 | 72.30 |
| | BBAVectors [59] | R-101 | ✓ | 88.35 | 79.96 | 50.69 | 62.18 | 78.43 | 78.98 | 87.94 | 90.85 | 83.58 | 84.35 | 54.13 | 60.24 | 65.22 | 64.28 | 55.70 | 72.32 |
| | DRN [14] | H-104 | ✓ | 89.71 | 82.34 | 47.22 | 64.10 | 76.22 | 74.43 | 85.84 | 90.57 | 86.18 | 84.89 | 57.65 | 61.93 | 69.30 | 69.63 | 58.48 | 73.23 |
| | PolarDet [60] | R-101 | ✓ | 89.65 | **87.07** | 48.14 | 70.97 | 78.53 | 80.34 | 87.45 | 90.76 | 85.63 | 86.87 | 61.64 | 70.32 | 71.92 | 73.09 | 67.15 | 76.64 |
| | RDD [61] | R-101 | ✓ | 89.15 | 83.92 | 52.51 | 73.06 | 77.81 | 79.00 | 87.08 | 90.62 | 86.72 | 87.15 | 63.96 | 70.29 | 76.98 | 75.79 | 72.15 | 77.75 |
| | GWD [5] | R-152 | ✓ | 89.06 | 84.32 | 55.33 | 77.53 | 76.95 | 70.28 | 83.95 | 89.75 | 84.51 | 86.06 | **73.47** | 67.77 | 72.60 | 75.76 | 74.17 | 77.43 |
| | KLD | R-50 | | 88.91 | 83.71 | 50.10 | 68.75 | 78.20 | 76.05 | 84.58 | 89.41 | 86.15 | 85.28 | 63.15 | 60.90 | 75.06 | 71.51 | 67.45 | 75.28 |
| | KLD | R-50 | ✓ | 88.91 | 85.23 | 53.64 | 81.23 | 78.20 | 76.99 | 84.58 | 89.50 | 86.84 | 86.38 | 71.69 | 68.06 | 75.95 | 72.23 | 75.42 | 78.32 |
| Refine-stage | CFC-Net [33] | R-101 | | 89.08 | 80.41 | 52.41 | 70.02 | 76.28 | 78.11 | 87.21 | 90.89 | 84.47 | 85.64 | 60.51 | 61.52 | 67.82 | 68.02 | 50.09 | 73.50 |
| | R$^3$Det [28] | R-152 | ✓ | 89.80 | 83.77 | 48.11 | 66.77 | 78.76 | 83.27 | 87.84 | 90.82 | 85.38 | 85.51 | 65.67 | 62.68 | 67.53 | 78.56 | 72.62 | 76.47 |
| | DAL [15] | R-50 | ✓ | 89.69 | 83.11 | 55.03 | 71.00 | 78.30 | 81.90 | 88.46 | 90.89 | 84.97 | **87.46** | 64.41 | 65.65 | 76.86 | 72.09 | 64.35 | 76.95 |
| | DCL [46] | R-152 | ✓ | 89.26 | 83.60 | 53.54 | 72.76 | 79.04 | 82.56 | 87.31 | 90.67 | 86.59 | 86.98 | 67.49 | 66.88 | 73.29 | 70.56 | 69.99 | 77.37 |
| | RIDet [34] | R-50 | ✓ | 89.31 | 80.77 | 54.07 | 76.38 | **79.81** | 81.99 | **89.13** | 90.72 | 83.58 | 87.22 | 64.42 | 67.56 | 78.08 | 79.17 | 62.07 | 77.62 |
| | S$^2$A-Net [13] | R-101 | ✓ | 89.28 | 84.11 | 56.95 | 79.21 | **80.18** | 82.93 | **89.21** | 90.86 | 84.66 | **87.61** | 71.66 | 68.23 | **78.58** | 78.20 | 65.55 | 79.15 |
| | R$^3$Det-GWD [5] | R-152 | ✓ | 89.66 | 84.99 | **59.26** | **82.19** | 78.97 | **84.83** | 87.70 | 90.21 | 86.54 | 86.85 | **73.04** | 67.56 | 76.92 | 79.22 | 74.92 | **80.19** |
| | R$^3$Det-KLD | R-50 | | 88.90 | 84.17 | 55.80 | 69.35 | 78.72 | 84.08 | 87.00 | 89.75 | 84.32 | 85.73 | 64.74 | 61.80 | 76.62 | 78.49 | 70.89 | 77.36 |
| | R$^3$Det-KLD | R-50 | ✓ | 89.90 | 84.91 | 59.21 | 78.74 | 78.82 | 83.95 | 87.41 | 89.89 | 86.63 | 86.69 | 70.47 | 70.87 | 76.96 | 79.40 | **78.62** | 80.17 |
| | R$^3$Det-KLD | R-152 | ✓ | 89.92 | 85.13 | 59.19 | **81.33** | 78.82 | 84.38 | 87.50 | 89.80 | 87.33 | 87.00 | 72.57 | **71.35** | 77.12 | 79.34 | **78.68** | **80.63** |

## 5  Discussions

**Limitations.** Despite the theoretical grounds and the promising experimental justifications, our method has an obvious limitation that it cannot be directly applied to quadrilateral detection [34, 45].

**Potential negative societal impacts.** Our findings provides a simple regression loss for high-precision rotation detection. However, our research may be applied to some sensitive fields, such as remote sensing, aviation, and unmanned aerial vehicles.

**Conclusion.** Departure from the vast existing literature in object detection, in this paper we have designed a new regression loss for rotation detection from scratch and consider the popular horizontal detection as its special case. Specifically, we calculate the KLD between the Gaussian distributions corresponding to the rotated bounding box as the regression loss, and we find that in the learning procedure guided by the KLD loss, the gradient of the parameters can be dynamically adjusted according to the characteristics of the object which is a desirable property for robust object detection, regardless its rotation, size and aspect ratio etc. We also proved that KLD has scale invariance, which is crucial for detection tasks. Interestingly, we have shown that KLD can be degenerated into the currently commonly used $l_n$-norm loss in the horizontal detection task. Extensive experimental results across different detectors and datasets show the effectiveness of our approach.

## Acknowledgments and Disclosure of Funding

This work was partly supported by National Key Research and Development Program of China (2018AAA0100704), Shanghai Municipal Science and Technology Major Project (2021SHZDZX0102) and NSFC (U20B2068, 72061127003, 61972250). Xue Yang was also partly supported by Wu Wen Jun Honorary Doctoral Scholarship, AI Institute, Shanghai Jiao Tong University, Shanghai, China.

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
