# Learning High-Precision Bounding Box for Rotated Object Detection via Kullback-Leibler Divergence Appendix

**Xue Yang**[1],[*] **Xiaojiang Yang**[1], **Jirui Yang**[2], **Qi Ming**[3], **Wentao Wang**[1], **Qi Tian**[4], **Junchi Yan**[1],[†]

[1]Department of Computer Science and Engineering,
MoE Key Lab of Artificial Intelligence, AI Institute, Shanghai Jiao Tong University
[2]University of Chinese Academy of Sciences
[3]Beijing Institute of Technology  [4]Huawei Inc.
{yangxue-2019-sjtu, yangxiaojiang, wwt117, yanjunchi}@sjtu.edu.cn
{yangjirui123, chaser.ming}@gmail.com tian.qi1@huawei.com

## A Proof of Scale Invariance of KLD

Suppose there are two Gaussian distributions, denoted as $\mathbf{X}_p \sim \mathcal{N}_p(\boldsymbol{\mu}_p, \boldsymbol{\Sigma}_p)$ and $\mathbf{X}_t \sim \mathcal{N}_t(\boldsymbol{\mu}_t, \boldsymbol{\Sigma}_t)$. Then, for a full-rank matrix $\mathbf{M}$, $|\mathbf{M}| \neq 0$, we have $\mathbf{X}_{p'} = \mathbf{M}\mathbf{X}_p \sim \mathcal{N}_p(\mathbf{M}\boldsymbol{\mu}_p, \mathbf{M}\boldsymbol{\Sigma}_p\mathbf{M}^\top)$, $\mathbf{X}_{t'} = \mathbf{M}\mathbf{X}_t \sim \mathcal{N}_t(\mathbf{M}\boldsymbol{\mu}_t, \mathbf{M}\boldsymbol{\Sigma}_t\mathbf{M}^\top)$, denoted as $\mathcal{N}_{p'}$ and $\mathcal{N}_{t'}$. The Kullback-Leibler Divergence (KLD) between $\mathcal{N}_{p'}$ and $\mathcal{N}_{t'}$ is:

$$
\begin{aligned}
\mathbf{D}_{kl}(\mathcal{N}_{p'}||\mathcal{N}_{t'}) =& \frac{1}{2}(\boldsymbol{\mu}_p - \boldsymbol{\mu}_t)^\top \mathbf{M}^\top (\mathbf{M}^\top)^{-1} \boldsymbol{\Sigma}_t^{-1} \mathbf{M}^{-1} \mathbf{M}(\boldsymbol{\mu}_p - \boldsymbol{\mu}_t) \\
&+ \frac{1}{2}\mathbf{Tr}\left((\mathbf{M}^\top)^{-1} \boldsymbol{\Sigma}_t^{-1} \mathbf{M}^{-1} \mathbf{M} \boldsymbol{\Sigma}_p \mathbf{M}^\top\right) \\
&+ \frac{1}{2}\ln\frac{|\mathbf{M}||\boldsymbol{\Sigma}_t||\mathbf{M}^\top|}{|\mathbf{M}||\boldsymbol{\Sigma}_p||\mathbf{M}^\top|} - 1 \\
=& \frac{1}{2}(\boldsymbol{\mu}_p - \boldsymbol{\mu}_t)^\top \boldsymbol{\Sigma}_t^{-1} (\boldsymbol{\mu}_p - \boldsymbol{\mu}_t) \\
&+ \frac{1}{2}\mathbf{Tr}\left(\mathbf{M}^\top (\mathbf{M}^\top)^{-1} \boldsymbol{\Sigma}_t^{-1} \mathbf{M}^{-1} \mathbf{M} \boldsymbol{\Sigma}_p\right) \\
&+ \frac{1}{2}\ln\frac{|\boldsymbol{\Sigma}_t|}{|\boldsymbol{\Sigma}_p|} - 1 \\
=& \mathbf{D}_{kl}(\mathcal{N}_p||\mathcal{N}_t)
\end{aligned}
\tag{1}
$$

Therefore, KLD has affine invariance. Especially when $\mathbf{M} = k\mathbf{I}$ ($\mathbf{I}$ denotes identity matrix), the scale invariance of KLD is proved.

## B Analysis of $\mathbf{D}_{kl}(\mathcal{N}_t||\mathcal{N}_p)$'s High-Precision Detection

The $\mathbf{D}_{kl}(\mathcal{N}_t||\mathcal{N}_p)$ between two 2-D Gaussian is:

$$
\mathbf{D}_{kl}(\mathcal{N}_t||\mathcal{N}_p) = \frac{1}{2}(\boldsymbol{\mu}_p - \boldsymbol{\mu}_t)^\top \boldsymbol{\Sigma}_p^{-1} (\boldsymbol{\mu}_p - \boldsymbol{\mu}_t) + \frac{1}{2}\mathbf{Tr}(\boldsymbol{\Sigma}_p^{-1}\boldsymbol{\Sigma}_t) + \frac{1}{2}\ln\frac{|\boldsymbol{\Sigma}_p|}{|\boldsymbol{\Sigma}_t|} - 1
\tag{2}
$$

---

[*]Part of the work was done during an internship at Huawei Inc.
[†]Correspondence author is Junchi Yan.

35th Conference on Neural Information Processing Systems (NeurIPS 2021).

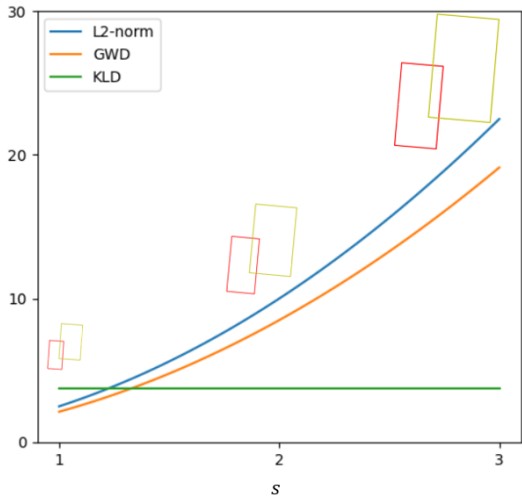

Figure 1: $L_2$-norm, GWD and KLD versus scaling factor.

each item of Eq. 2 can be expressed as

$$(\boldsymbol{\mu}_p - \boldsymbol{\mu}_t)^\top \boldsymbol{\Sigma}_p^{-1}(\boldsymbol{\mu}_p - \boldsymbol{\mu}_t) = \frac{4\left(\Delta x \cos\theta_p + \Delta y \sin\theta_p\right)^2}{w_p^2} + \frac{4\left(\Delta y \cos\theta_p - \Delta x \sin\theta_p\right)^2}{h_p^2} \tag{3}$$

$$\mathbf{Tr}(\boldsymbol{\Sigma}_p^{-1}\boldsymbol{\Sigma}_t) = \frac{h_t^2}{w_p^2}\sin^2\Delta\theta + \frac{w_t^2}{h_p^2}\sin^2\Delta\theta + \frac{h_t^2}{h_p^2}\cos^2\Delta\theta + \frac{w_t^2}{w_p^2}\cos^2\Delta\theta \tag{4}$$

$$\ln\frac{|\boldsymbol{\Sigma}_p|}{|\boldsymbol{\Sigma}_t|} = \ln\frac{h_p^2}{h_t^2} + \ln\frac{w_p^2}{w_t^2} \tag{5}$$

where $\Delta x = x_p - x_t, \Delta y = y_p - y_t, \Delta\theta = \theta_p - \theta_t$.

For the parameter $\mu_p$, we have

$$\frac{\partial f_{kl}(\mu_p)}{\partial \mu_p} = \begin{pmatrix} \frac{4}{w_p^2}(\Delta x \cos\theta_p + \Delta y \sin\theta_p)\cos\theta_p + \frac{4}{h_p^2}(-\Delta x \sin\theta_p + \Delta y \cos\theta_p)(-\sin\theta_p) \\ \frac{4}{w_p^2}(\Delta x \cos\theta_p + \Delta y \sin\theta_p)\sin\theta_p + \frac{4}{h_p^2}(-\Delta x \sin\theta_p + \Delta y \cos\theta_p)\cos\theta_p \end{pmatrix} \tag{6}$$

It is assumed that except for $\mu_p$, other parameters have been optimized to the best. In other words, $h_p = h_t$, $w_p = w_t$, and $\theta_p = \theta_t$. Without loss of generality, we set $\theta_t = 0°$, then

$$\frac{\partial f_{kl}(\mu_p)}{\partial \mu_p} = \left(\frac{4}{w_t^2}\Delta x, \frac{4}{h_t^2}\Delta y\right)^\top \tag{7}$$

The weights $1/w_t^2$ and $1/h_t^2$ will make the model dynamically adjust the optimization of the object position according to the scale.

For $h_p$ and $w_p$, we have

$$\begin{aligned}
\frac{\partial f_{kl}(\boldsymbol{\Sigma}_p)}{\partial \ln h_p} &= 1 - \frac{4\left(\Delta y \cos\theta_p - \Delta x \sin\theta_p\right)^2}{h_p^2} - \frac{w_t^2}{h_p^2}\sin^2\Delta\theta - \frac{h_t^2}{h_p^2}\cos^2\Delta\theta \\
\frac{\partial f_{kl}(\boldsymbol{\Sigma}_p)}{\partial \ln w_p} &= 1 - \frac{4\left(\Delta x \cos\theta_p + \Delta y \sin\theta_p\right)^2}{w_p^2} - \frac{h_t^2}{w_p^2}\sin^2\Delta\theta - \frac{w_t^2}{w_p^2}\cos^2\Delta\theta
\end{aligned} \tag{8}$$

Similarly, suppose $\Delta x = \Delta y = \Delta\theta = 0$, $\frac{\partial f_{kl}(\boldsymbol{\Sigma}_p)}{\partial \ln h_p} = 1 - \frac{h_t^2}{h_p^2}$, $\frac{\partial f_{kl}(\boldsymbol{\Sigma}_p)}{\partial \ln w_p} = 1 - \frac{w_t^2}{w_p^2}$, which means that the smaller targeted height or width leads to heavier penalty on its matching loss. This is desirable, as smaller height or width needs higher matching precision.

Similarly, suppose $\Delta x = \Delta y = \Delta = 0$ and $w_p = w_t, h_p = h_t$, we have

$$\frac{\partial f_{kl}(\boldsymbol{\Sigma}_p)}{\partial \theta_p} = \left(\frac{h_t^2}{w_t^2} + \frac{w_t^2}{h_t^2} - 2\right)\sin 2\Delta\theta \geq \sin 2\Delta\theta \tag{9}$$

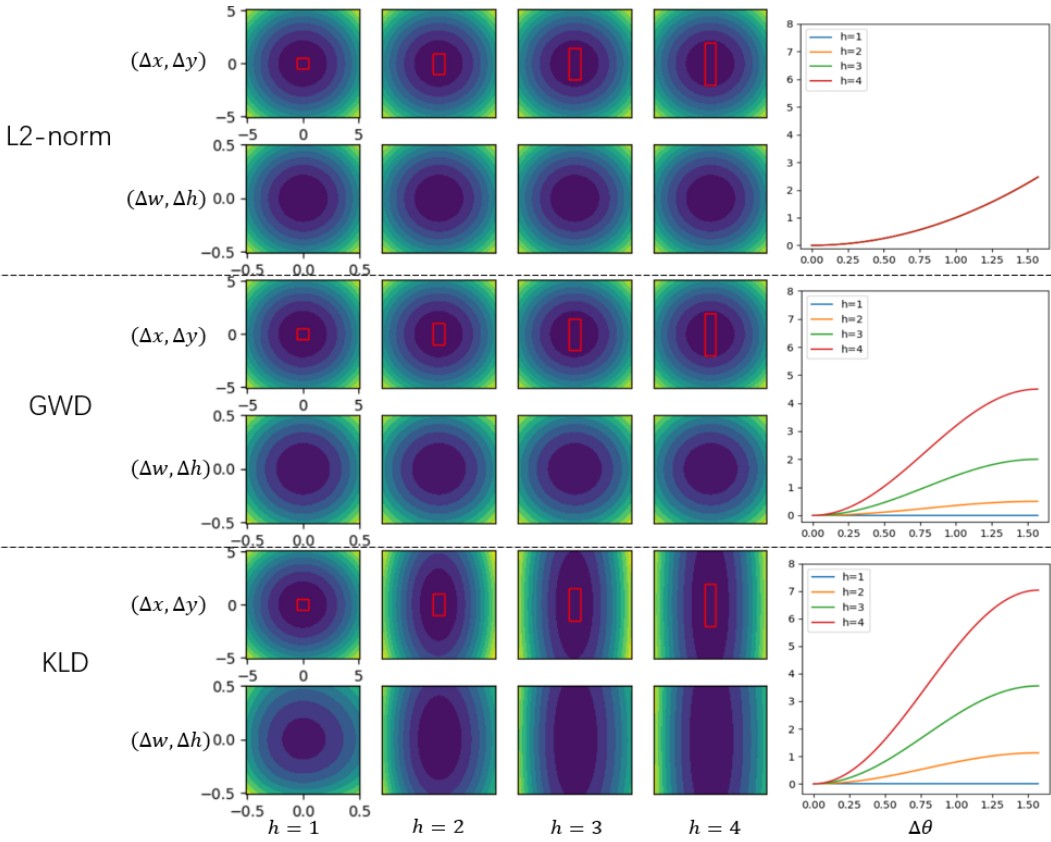

Figure 2: $L_2$-norm, GWD and KLD versus parameters when the targeted height varies.

the condition for the equality sign is $h_t = w_t$. This shows that the larger the aspect ratio of the object, the model will pay more attention to the optimization of the angle.

Compared with $\mathbf{D}_{kl}(\mathcal{N}_p||\mathcal{N}_t)$, $\mathbf{D}_{kl}(\mathcal{N}_t||\mathcal{N}_p)$ has a similar gradient optimization strategy. The difference is that the relationship between the parameters of $\mathbf{D}_{kl}(\mathcal{N}_t||\mathcal{N}_p)$ is tighter.

## C The Visualization of KLD's Advantages

This section aims to visually show the advantages of KLD, including its scale invariance and ability of high-precision detection. To this end, we compare KLD with $L_2$-norm and GWD, pointing out that these advantages are characteristics of KLD.

To visualize the scale invariance of KLD, we consider the KLD of two given boxes, and investigate the variation of KLD when the two boxes are enlarged with a scaling factor $s$. Specifically, the parameters of the two boxes are $(0, 0, s, 2s, 5°)$ and $(s, s, 1.1s, 2.2s, 5°)$, respectively. As shown in Figure 1, the value of KLD is invariant to the scaling factor $s$. Compareed with this, the values of $L_2$-norm and GWD change when $s$ increases, and hence they have no advantage of scale invariance. Note that IoU is also invariant to the scaling of boxes, and hence to some degree, KLD is a better substitute of IoU than $L_2$-norm and GWD.

The ability of high-precision detection of KLD means it do well in object detection with large aspect ratio. Specifically, for bounding box with larger aspect ratio, KLD gives heavier penalties to matching of shorter edge's length and the center point's position along the shorter edge's direction, as well as the matching of angle. These characteristics are desirable, as when matching bounding box with large aspect ratio, IoU is intuitively sensitive to the shorter edge's length, the center point's position along the shorter edge's direction and the angle. To visualize these characteristics of KLD, we consider a target box with $x = 0$, $y = 0$, $w = 1$, $\theta = 0$, and set $h = \{1, 2, 3, 4\}$ to control the aspect ratio, and plot KLD versus the variation of parameters. $L_2$-norm and GWD are also included for comparisons. As shown in Figure 2, when $h$ increases, KLD is more sensitive to the variation of $x$, $w$ and $\theta$,

meaning it has desirable advantages for object detection with large aspect ratio. Compared with this, both $L_2$-norm and GWD pay no more attention to the matching of $x$ and $w$ when $h$ increases, and $L_2$-norm is even unchanged when the difference of angle $\Delta\theta$ is fixed.