# OpenReview forum: "Learning High-Precision Bounding Box for Rotated Object Detection via Kullback-Leibler Divergence"
_NeurIPS.cc/2021/Conference — NeurIPS 2021 Poster_

### Official Review · Reviewer_7RYn · 2021-07-05

**Rating:** 2
**Confidence:** 4

**Summary:**

The paper proposes an algorithm that can detect slender objects,
like text in images or boats in aerial images. The work mainly focuses on
the bounding box regression task and includes a parameter for box rotation.
Bounding boxes are represented as Gaussian functions instead of axis-aligned
boxes. The scale invariant KL-divergence loss is applied to the regression task.
An analysis of the loss function proclaims that the computation of gradients,
during optimization, are adjusted with respect to the bounding box parameters.

**Limitations And Societal Impact:**

It is not clear why the listed fields (line
288) are sensitive. An example for one of the fields could give a more complete
understanding of the negative impacts.

**Main Review:**

The paper is not reproducible, motivations and explanations
are incomplete. This paper is not yet ready for NeurIPS 2021.

1. The paper gives a vague description of the problem, why is it important
and what are the challenges. There is insufficient motivation. A clear
description of why and the final end-use should be included. In this end-use, why are rotated bounding boxes better compared to another high-
precision object representation, for example segmentation.
2. A clear motivation of why it is more correct to directly predict bounding
box parameters, rather then parameter offsets, is not given. Both can be
equally good depending on prior assumptions.
3. Line 229, what kind of data augmentation?
4. It is a strong claim to say that horizontal bounding boxes are a special
case of something else without any references. Equations (4) and (16) are
not very similar. There are large differences, more terms and also different
norms.
5. Missing references. Claims: lines 102, 104-105, 141-144, 212. Terms: lines
275, 276, 279.
6. Which methods do you refer to when you say they are not scale invariant?
In SSD, the offsets are very near a standard normal distribution. It is
unclear why a scale invariant loss versus a simple scaling of the loss is
beneficial.
7. You want to convey that the KL-loss dynamically adjust gradients w.r.t
width and height. However, the details do
not clearly explain this statement. E.g. is function f (13) defined here or explained somewhere else?
8. Usually L1-norm is better when there are outliers. Here outliers are allowed to contribute more to the loss, due to the quadratic terms in the KL-divergence loss. From the ablation this is solved by (18). This extra normalization questions if the KL-divergence is actually contributing
or simply produces noise in the results. This loss would have been better motivated if Table 6 was extended with more of the state-of-the-art
detectors, where axis aligned boxes are used.
9. References [6,7,8,9,10] do not include an extra log term in the bounding
box regression. In (4) and extra log term is added.
10. Line 63, not well motivated or explained.
11. Line 191, is f the same as in (13), (14), (15)?
12. Needs proof reading. E.g. line 38, (13), line 143, line 194, line 279.

Update after rebuttal:
several questions have been addressed in the rebuttal. However, question 8 is not sufficiently accurately addressed. Also: what does "We will discuss above in detail in the final version." mean concretely.
Given the responses from the reviewers, I have raised my assessment, but the paper is still not of sufficient quality for acceptance.

Update after discussion with the authors:
The disrespectful tone during the discussion is unprofessional and not suitable for a scientific top tier conference. The rating is changed accordingly.

**Time Spent Reviewing:**

18

---

> ### Author Response · Authors · 2021-08-05
> **Rebuttal to Reviewer 7RYn**
>
> Thank you for your detailed comments.
>
> **Q1**. The paper is not reproducible, the description of the problem is vague, and the motivation is insufficient.
>
> **A1**.
>
> **Repeatability**: We have already submitted the code in supplementary material, so the paper is reproducible.
>
> **Problem description**: Rotation detection is often applied to aerial images, scene text, human faces, 3D objects, retail scenes, etc. Compared with horizontal object detection, the research on rotation detection is at a relatively early stage, and there are many unsolved problems that need to be solved, **such as dense arrangement, high precision, and large aspect ratio**. **Since segmentation requires a lot of labeling time and cost, the representation of the bounding box is more cost-effective**, and this representation can basically meet the needs of most scenes.  Based on this representation, this paper explores how to make the model predict a more accurate detection bounding box. There are many excellent works for similar purposes [2-5], including horizontal detection and rotation detection
>
> **Motivation**: The recent work (GWD) [1] analyzes that the introduction of N (µ, Σ) can solve the inconsistency between metric and loss, boundary discontinuity and square-like problem.  **On this basis, we further studies how to design high-precision detection regression loss through new parameter space.** Our view is that the self-modulated mechanism is positively correlated with the final high-precision performance. Through the gradient analysis of each parameter in KLD, we further find that the **self-modulated optimization mechanism** of KLD greatly promotes the improvement of high-precision detection, much better than GWD and baseline. More importantly, we have theoretically shown that KLD is **scale invariant** for detection, while GWD is not.
>
> **We will discuss above in detail in the final version.**
>
> [1] X. Yang, J. Yan, M. Qi, W. Wang, Z. Xiaopeng, and T. Qi, “Rethinking rotated object detection with Gaussian Wasserstein distance loss,” in International Conference on Machine Learning, 2021.
>
> [2] Chen, K., Pang, J., Wang, J., Xiong, Y., Li, X., Sun, S., ... & Lin, D. (2019). Hybrid task cascade for instance segmentation. In Proceedings of the IEEE/CVF Conference on Computer Vision and Pattern Recognition (pp. 4974-4983).
>
> [3] Cai, Z., & Vasconcelos, N. (2018). Cascade r-cnn: Delving into high quality object detection. In Proceedings of the IEEE conference on computer vision and pattern recognition (pp. 6154-6162).
>
> [4] Zhang, S., Wen, L., Bian, X., Lei, Z., & Li, S. Z. (2018). Single-shot refinement neural network for object detection. In Proceedings of the IEEE conference on computer vision and pattern recognition (pp. 4203-4212).
>
> [5] Yang, X., Yan, J., Feng, Z., & He, T. (2021, May). R3Det: Refined Single-Stage Detector with Feature Refinement for Rotating Object. In Proceedings of the AAAI Conference on Artificial Intelligence (Vol. 35, No. 4, pp. 3163-3171).
>
> **Q2**. About the prediction of the model.
>
> **A2**. We just modified the loss function of the baseline, and **we still predict the parameter offsets**. The process is as follows:
>
> predict offset $(\Delta t_{x}, \Delta t_{y}, \Delta t_{w}, \Delta t_{h}, \Delta t_{\theta})$ -> decode prediction box -> convert prediction box and gt into Gaussian distribution -> calculate KLD of two Gaussian distributions.
>
> **Q3**. Line 229, what kind of data augmentation?
>
> **A3**. As described in the caption of Tab. 3, data augmentation includes **random rotation, flipping, graying**. We will add it in line 229.
>
> **Q4**. Why horizontal bounding boxes are a special case. Equations (4) and (16) are not very similar.
>
> **A4**. According to the definition of the rotating bounding box $(x,y,w,h,\theta)$, when $\theta=0$, it becomes a horizontal box. According to lines 186-187, Eq. (4) and Eq. (16) are indeed slightly different, which is why KLD performs better in Tab. 6.
>
> **Q5**. Missing references.
>
> **A5**. We will add relevant references.
>
> **Q6**. About scale invariant.
>
> **A6**. **Smooth L1 and GWD are scale invariant.** Many detectors, such as SSD and Faster RCNN, will normalize the input parameters in advance like Eq. (4) before using Smooth L1, which also illustrates the importance of scale invariant for detection. However, **KLD itself is scale invariant, so no normalization operation is required.** This paper mainly wants to compare KLD with GWD, which is neither scale-scaling nor scale-invariant from this aspect, as one of the reasons why KLD is better.
>
> **Q7**. Details on the dynamic adjustment of the gradients w.r.t width and height.
>
> **A7**. In Eq. (14) and lines 166-186. The gradients of width and height will be dynamically adjusted according to the $\Delta \theta$.
>
> **Q8**. This extra normalization questions if the KL-divergence is actually contributing or simply produces noise in the results.
>
> **A8**. The reason you describe is exactly why we use Eq. (18). **If the normalization operation is not used, the loss value of GWD and KLD are too large, resulting in NAN in training.** In order to further prove that KLD is indeed effective, the relevant experimental results are as follows:
>
> **HRSC2016 (Hmean50/Hmean75/Hmean50:95):**
>
> Smooth-L1: 84.8/48.42/47.76
>
> $1-\frac{1}{1+Smooth-L1}$: 81.28/53.39/47.64
>
> $1-\frac{1}{1+log(Smooth-L1+1)}$: 78.99/43.12/43.47
>
> GWD: 85.56/60.31/52.89
>
> KLD: **87.45/72.39/57.80**
>
> **DOTA-v1.0 (AP):**
>
> Smooth-L1: 65.73
>
> $1-\frac{1}{1+Smooth-L1}$: 64.95
>
> GWD: 68.93
>
> KLD: **71.28**
>
> **The above experimental results prove that the effectiveness of KLD does not come from Eq. (18).**
>
> **COCO (AP/AP50/AP75/APs/APm/APl):**
>
> FCOS+IoU: 36.6/56.0/38.8/21.0/40.6/47.0
>
> FCOS+KLD: **36.8/56.3/39.1/21.7/40.8/47.5**
>
> **Similar to the conclusion of Tab. 6 in the paper, KLD is not inferior to IoU/GIoU loss.**
>
> **Q9**.  Error of Eq. (4)
>
> **A9**.
> Thank you for your correction. Eq. (4) should be:
> \begin{equation}
> 	L_{reg} = l_{n}\text{-norm}\left(\Delta t_{x}, \Delta t_{y}, \Delta t_{w}, \Delta t_{h}, \Delta t_{\theta}\right)
> \end{equation}
> where $\Delta t_{x} = t_{x}^{p} - t_{x}^{t} = \frac{\Delta x}{w_{a}}$, $\Delta t_{y} = t_{y}^{p} - t_{y}^{t} = \frac{\Delta y}{h_{a}}$, $\Delta t_{w} = t_{w}^{p} - t_{w}^{t} = \ln(w_{p}/w_{t})$, $\Delta t_{h} = t_{h}^{p} - t_{h}^{t} = \ln(h_{p}/h_{t})$, and $\Delta t_{\theta} = t_{\theta}^{p} - t_{\theta}^{t} = \Delta \theta$.
>
> Eq. (16) should be:
> \begin{equation}
>     2l_{2}\text{-norm}(\Delta t_{x}, \Delta t_{y})+l_{1}\text{-norm}(\Delta t_{w}, \Delta t_{h})+\frac{1}{2}l_{2}\text{-norm}(\frac{1}{\Delta t_{w}}, \frac{1}{\Delta t_{h}}) -1
> \end{equation}
>
> **Q10**. Line 63, not well motivated or explained.
>
> **A10**. For common application scenarios of rotation detection (such as aerial images), **the scale of the object varies greatly**, often including objects less than 10 pixels (such as cars, yachts), and also very large objects (such as sports fields, aircraft carriers). In addition, there are often objects that are extremely long on one side and extremely short on the other (such as harbor, bridge). Therefore, scale invariance is very important for rotation detection.
>
> **Q11**. Line 191, is f the same as in (13), (14), (15)?
>
> **A11**. It is not the same. It is more accurate to replace $f_{kf}$ in formula (13)-(15) with $\mathbf{D}_{kl}$, thanks.
>
> **Q12**. Needs proof reading. E.g. line 38, (13), line 143, line 194, line 279.
>
> **A12**. I will correct it on the final version, thanks.
>
> **Q13**. Why the listed fields (line 288) are sensitive.
>
> **A13**. Remote sensing scenarios often involve military objects, so we think they are sensitive.
>
> **We hope this response could help address your concerns, and wish to receive your further feedback soon.**

---

> ### Author Response · Authors · 2021-08-26
> **Rebuttal to Reviewer 7RYn (Round 2)**
>
> **Q1:** Question 8 is not sufficiently accurately addressed
>
> **A1** We have added relevant experiments to eliminate the interference of Eq. (18) according to your suggestion to prove the effectiveness of KLD.
>
> - IThe normalization (Eq. 18) is proposed by GWD. Therefore, in order to maintain the fairness of the comparison, we perform the same operation, and the results show that KLD is better than GWD.
>
> - We have added two sets of baseline experiments with and without normalization operations. The results show that KLD still has better performance, and the normalization operation does not improve the performance of the baseline.
>
> We have also added verification experiments on more horizontal detectors (FCOS). May I ask what else we need to do to eliminate your confusion ("question 8 is not sufficiently accurately addressed").
>
> **Q2:** Also: what does "We will discuss above in detail in the final version." mean concretely.
>
> **A2:** Sorry for misleading you. What we mean is that the paper will be revised based on your comments.

---

> > ### Comment · Reviewer_7RYn · 2021-08-26
> > **Question 8**
> >
> > My original comment "This loss would have been better motivated if
> > Table 6 was extended with more of the state-of-the-art
> > detectors, where axis aligned boxes are used" has been addressed by
> > additional experiments on FCOS, but does not support the authors' claim
> > on line 276. Also, the new results are similar to the previous one and
> > do not show a significant difference. I stay with my final assessment.

---

> > > ### Author Response · Authors · 2021-08-26
> > > **Rebuttal to Reviewer 7RYn**
> > >
> > > Thanks for your quick reply. The following are my views and explanations on the effectiveness of KLD in horizontal detection:
> > >
> > > - The ground truth for rotation detection is the minimum circumscribed rectangle, which means that ground truth can well reflect the true scale and direction information of the object. The "special horizontal case" described in this paper also meets the above requirements, the horizontal circumscribed rectangle is equal to the minimum circumscribed rectangle at this time. Although the ground truth of the COCO is a horizontal box, it is not the minimum circumscribed rectangle, which means that it loses the direction information and accurate scale information of the object. For example, a baseball bat placed obliquely in the image, the height and width of its horizontal circumscribed rectangle do not represent the height and width of the object itself. This causes that when KLD is applied to the COCO, the optimization mechanism of KLD that dynamically adjusts the angle gradient according to the aspect ratio is meaningless, which affects the improvement of the final performance. In general, this is a defect in the dataset annotation itself, not that KLD is not good enough.
> > >
> > > - When $\theta=0$, the first four items of KLD are almost the same as the commonly used $L_{n}$ regression loss. The main difference is that there are additional two items and different normalization. The above difference explains why the KLD is better than the commonly used $L_{n}$ loss, comparable to or even better than IoU/GIoU loss. This is also why GWD[1] is proposed to approximate the non-differentiable rotated IoU loss, but the scale invariance of KLD makes it more suitable than GWD.
> > >
> > > The above two points explain why the performance advantage of KLD on the COCO dataset is not significant. We think reviewers should be aware that the research topic of this paper is rotation detection. As Reviewer eGtr said: "I do not consider this is a major issue as this works very well rotated object detection which is also valuable." In addition, if the reviewer gives a low score to reject the manuscript because KLD is not outstanding enough on the horizontal dataset (e.g. COCO), we think it is not objective.
> > >
> > > We hope the reviewers can consider my explanation and look forward to your feedback again.
> > >
> > > [1] X. Yang, J. Yan, M. Qi, W. Wang, Z. Xiaopeng, and T. Qi, “Rethinking rotated object detection with Gaussian Wasserstein distance loss,” in International Conference on Machine Learning, 2021.

---

> > > > ### Comment · Reviewer_7RYn · 2021-08-27
> > > > **not just about COCO**
> > > >
> > > > The low score is based on several points listed in the original review. The COCO results are not the only issue that remains even after the rebuttal.
> > > > The COCO results have been lifted in the discussion as particularly problematic as convincing results on COCO are required for the authors' line of arguments. Thus, non-significant improvement of results on COCO jeopardizes the premise and leaves the conclusions uncertain.
> > > > With this uncertainty in mind, I cannot rate differently than to reject the paper.

---

> ### Author Response · Authors · 2021-08-27
> **Rebuttal to Reviewer 7RYn (Round 3)**
>
> Thank you for your reply.
>
> Indeed, the conclusion on line 276 ( This provides a strong support for our original idea: to design a rotation regression loss with high-precision detection potential.) in the paper is inaccurate. What I originally meant was that even in the task of horizontal detection with inaccurate ground truth, KLD can still bring about a performance gain similar to IoU/GIoU loss compared to $L_{n}$. Therefore, KLD is an excellent **rotation regression loss**, rather than saying that KLD is an excellent loss in horizontal detection. I will reorganize to express this meaning more accurately. Furthermore, the horizontal bounding box labeling of COCO and the horizontal situation of rotation detection are not the same thing, the reason has been explained in the last answer. Therefore, COCO's experiment is of little significance to verify the effectiveness of KLD.
>
> As for COCO is not the only reason for rejection. We think it is necessary for reviewers to raise the unresolved or concerned issues again in each discussion session, so that we can respond more comprehensively instead of thinking that the unmentioned issues have been resolved. We think this is also the meaning of rolling discussion: even if the article is eventually rejected, it can be improved next time. Of course, for similar issues such as missing references (Q1, Q3, Q5, Q10, Q12), we can only guarantee to make corresponding supplements in the follow-up.
>
> We very much hope to continue to have an in-depth discussion with you on the core deficiencies of the paper.

---

> > ### Comment · Reviewer_7RYn · 2021-09-14
> > **your other responses**
> >
> > Similar to the paper, most responses leave room for interpretation, which is however not what I expect for the rebuttal.
> >
> > For instance, my issue 4 was "It is a strong claim to say that horizontal bounding boxes are a special case of something else without any references. Equations (4) and (16) are not very similar. There are large differences, more terms and also different norms."
> >
> > The response was "According to the definition of the rotating bounding box , when , it becomes a horizontal box. According to lines 186-187, Eq. (4) and Eq. (16) are indeed slightly different, which is why KLD performs better in Tab. 6."
> >
> > The last statement is irrelevant to answer the question. The authors agree that (4) differs from (16), which renders the original claim incorrect.

---

> > > ### Author Response · Authors · 2021-09-14
> > > **Rebuttal to Reviewer 7RYn**
> > >
> > > We think that "the horizontal box ($\theta$=0) is a special case of the rotating box" is a very intuitive conclusion, and can be intuitively drawn from the definition of the rotating bounding box [1-2]. We think that there is almost no work to prove such an intuitive conclusion, so we are puzzled why this needs references to demonstrate.
> > > We will further search for relevant materials, and if there are relevant documents, we will definitely cite them.
> > >
> > > [1] X. Yang, J. Yan, M. Qi, W. Wang, Z. Xiaopeng, and T. Qi, “Rethinking rotated object detection with Gaussian Wasserstein distance loss,” in International Conference on Machine Learning, 2021.
> > >
> > > [2] Yang, X., & Yan, J. Arbitrary-oriented object detection with circular smooth label. In European Conference on Computer Vision (pp. 677-694). Springer, Cham, 2020.
> > >
> > >
> > > For Eq. (4) and (16), the original statement in line 186-187 of our article is: where the **first two terms** of Eq. 16 are very similar to Eq. 4. Therefore, question 4 is taken out of context.
> > > If you are familiar with general object detection methods (such as Faster RCNN), the commonly used Ln loss in the Eq. (4) is Smooth L1. Smooth L1 is a combined function of L2 and L1, so comparing the first two items of Eq. (16) with Eq. (4), we point out that they are similar. In the actual application process, L1, L2 and Smooth L1 are all commonly used choices, so what we write in Eq. (4) is a more general Ln. The "similar" we describe means that they are similar in nature, which does not negate the influence of the weight and the form of Ln used. These slight differences plus the extra terms of Eq. (16) make the final result slightly different, that is, KLD performs better.
> > >
> > > **Eq. (4):**
> > > \begin{equation}
> > > 	L_{reg} = l_{n}\text{-norm}\left(\Delta t_{x}, \Delta t_{y}, \Delta t_{w}, \Delta t_{h}, \Delta t_{\theta}\right)
> > > \end{equation}
> > > where $\Delta t_{x} = t_{x}^{p} - t_{x}^{t} = \frac{\Delta x}{w_{a}}$, $\Delta t_{y} = t_{y}^{p} - t_{y}^{t} = \frac{\Delta y}{h_{a}}$, $\Delta t_{w} = t_{w}^{p} - t_{w}^{t} = \ln(w_{p}/w_{t})$, $\Delta t_{h} = t_{h}^{p} - t_{h}^{t} = \ln(h_{p}/h_{t})$, and $\Delta t_{\theta} = t_{\theta}^{p} - t_{\theta}^{t} = \Delta \theta$.
> > >
> > > **Eq. (16):**
> > > \begin{equation}
> > >     2l_{2}\text{-norm}(\Delta t_{x}, \Delta t_{y})+l_{1}\text{-norm}(\Delta t_{w}, \Delta t_{h})+\frac{1}{2}l_{2}\text{-norm}(\frac{1}{\Delta t_{w}}, \frac{1}{\Delta t_{h}}) -1
> > > \end{equation}

---

> > > > ### Comment · Reviewer_7RYn · 2021-09-14
> > > > **Inaccuracies in writing cannot be blamed on the reader**
> > > >
> > > > It is the task of the authors to formulate an accurate description of what they want to say. Implicit similarities or parallels as used by the authors do not meet this requirement.
> > > >
> > > > The rebuttal window is closed and further replies are not to be considered (I learned from another discussion).
> > > >
> > > > This manuscript does not meet the high standard of NeurIPS as argued earlier.

---

> > > > > ### Author Response · Authors · 2021-09-14
> > > > > **Rebuttal to Reviewer 7RYn**
> > > > >
> > > > > First of all, what we need to state is that this article is improved based on many classic methods, so readers need to have a certain foundation, such as understanding that the regression loss of Faster RCNN is smooth l1. This article does not have such a large space to introduce these common sense things. If the reviewer is not familiar with the content, we believe that the reviewer needs to reflect on whether he/her can be qualified for the review work in this field.
> > > > >
> > > > > In addition, regarding the question of Eq. (4) and (16), it is obvious that the reviewer did not read clearly. How can this blame the author instead.
> > > > >
> > > > > We acknowledge that there are issues in the article, and we will carefully revise them. However, we do not approve of reviewer‘s irresponsible behavior.

---

### Official Review · Reviewer_eGtr · 2021-07-15

**Rating:** 7
**Confidence:** 4

**Summary:**

This paper focuses on learning high precision bounding boxes for rotated object detections. Instead of directly predicting the size of the box, they turn the box into a gaussian distribution and use the KL divergence between the gaussian distributions as the regression loss. They show that KLD is modulated where the optimization of one parameter will be affected by other hyper-parameters. For example, the gradients for the orientation involve the width and height of a box. Experiments show that the proposed loss function significantly improve the performance of a detector on different aerial image datasets and scene text datasets.

**Limitations And Societal Impact:**

Yes.

**Main Review:**

The main contribution of this paper is that it shows why KLD is a better option than existing losses to train a network for rotated objects both analytically and empirically. Their results also generalize well across different datasets. Although this paper does not propose any new loss or new formulation of a task, this shows a strong result for future works on rotated object detection. Furthermore, this paper does a good job on explaining everything very clearly. It is very easy to follow. Overall, I think this is a good paper.

One downside of this approach is that it does not have much advantage on horizontal object detection such COCO as shown in table 6, which limits its practicality, considering that some corner-based detectors [1] already perform well at high IoU level on COCO. Having said that, I do not consider this is a major issue as this works very well rotated object detection which is also valuable.

The authors show that they are able to achieve a higher performance with a smaller network. So I would suggest the authors to include the inference time of their methods and compare it with other works.

1. Hei Law, Jia Deng: CornerNet: Detecting Objects as Paired Keypoints. Int. J. Comput. Vis. 128(3): 642-656 (2020)

**Time Spent Reviewing:**

4

---

> ### Author Response · Authors · 2021-08-05
> **Rebuttal to Reviewer eGtr**
>
> Thank you for your appreciation.
>
> **Q1**. KLD does not have much advantage on horizontal object detection.
>
> **A1**. The gt for rotation detection is the minimum bounding rectangle, which contains as little background area as possible, and the horizontal special cases described in the paper also meet the above requirements. Although the gt of the COCO is a horizontal box, it is not the minimum bounding rectangle, so the experimental results will be affected to a certain extent.
>
> **Q2**. Inference time.
>
> **A2**. This paper only proposes a new loss function, the process is as follows:
>
> predict offset $(\Delta t_{x}, \Delta t_{y}, \Delta t_{w}, \Delta t_{h}, \Delta t_{\theta})$ -> decode prediction box -> convert prediction box and gt into Gaussian distribution -> calculate KLD of two Gaussian distributions.
>
> Compared with the baseline, we only modify the loss function, which does not change the parameters of the original network, **so the inference time remains unchanged**.
>
> **We hope this response could help address your concerns, and wish to receive your further feedback soon.**

---

### Official Review · Reviewer_MV1m · 2021-07-17

**Rating:** 6
**Confidence:** 4

**Summary:**

This paper presents a new design of the regression loss function for oriented object detection. When modeling the rotated bounding boxes as 2D Gaussian distributions, the authors introduced the Kullback-Leibler Divergence for oriented bounding boxes regression as an alternative to the commonly-used Smooth-l1 loss function. The experimental results show that the proposed loss function could obtain better performance on several benchmarks.

**Limitations And Societal Impact:**

The authors have adequately addressed the limitations and potential negative societal impact of their work.

**Main Review:**

- This paper extended the idea of modeling the oriented bounding boxes as Gaussian distributions [1]. The main difference between this paper and [1] is using the KL Divergence instead of the Gaussian Wasserstein Distance as the regression loss function for the learning of oriented bounding boxes. Although better performance could be obtained, this paper did not present insights for the reason why using KL Divergence.  Accordingly, this paper is actually another attempt for finding a better-performing distance function under the framework of Gaussian modeling of oriented object detection. Without further exploiting the insights behind the proposed KLD loss, the main contributions claimed about the novelty of the proposed loss function by this paper are not that significant.

- The authors discussed the **self-modulated mechanism** of the KLD loss, however, it is only a trivial observation. As the final evaluation metric is based on the IoU computation between two bounding boxes in the 2D Euclidean space, the more strict requirement for the objects with a larger aspect ratio should be taken into account for the sake of better IoU. A feasible solution is improving the weight for learning large aspect-ratio objects.

- From another perspective, the proposed KLD loss couples the orientation and the width-height estimation together. As shown in equation (10), the residual in $x$ and $y$ directions are rotated by a rotation matrix $R(\theta) \in \text{SO}(2)$, which is actually a conversion from the polar coordinate system to the Euclidean one.

- The KLD loss only addressed the inaccurate angular regression of the **oriented** object detection for the larger aspect-ratio objects. For the detection of the horizontal bounding boxes, the problem of inaccurate regression for the width and height still remains.

## References
[1] X. Yang, J. Yan, M. Qi, W. Wang, Z. Xiaopeng, and T. Qi, “Rethinking rotated object detection with Gaussian Wasserstein distance loss,” in International Conference on Machine Learning, 2021.

**Time Spent Reviewing:**

3

---

> ### Author Response · Authors · 2021-08-05
> **Rebuttal to Reviewer MV1m**
>
> Thanks for your comment.
>
> **Q1.** Insights of KLD.
>
> **A1**. GWD[1] has explained the advantages of the framework based on Gaussian distribution modeling, such as overcoming the boundary discontinuity and square-like detection problems. This paper is based on the Gaussian distribution modeling framework to further explore which distribution measurement method is more suitable for rotation detection, especially compared to GWD. Specially, we present insights behind the proposed KLD in Sec. 3, include:
> i) **More advanced parameter optimization mechanism than GWD, proved by gradient analysis;**
> ii) **Compared with GWD, KLD is proven to have the advantage of scale invariance.**
>
> **Q2**. Novity of self-modulated mechanism.
>
> **A2**. For object detection, especially rotation detection, the optimization mechanism of parameters is essential. This paper uses **gradient analysis (Sec. 3)**, **gradient landscape (Appendix)**, **visualization of detection result (Fig. 2)**, **high-precision indicators comparison (Tab. 3-5)** to prove that KLD has an advanced parameter optimization mechanism, which mainly includes **adjusting the gradient of the center point offset according to the scale, and dynamically adjusting the gradient of the angle parameter according to the aspect ratio. This is definitely not a trivial observation but sufficient proof.** In contrast, adjusting the weights according to the aspect ratio is too simple. As mentioned in lines 106-107, a lot of tuning experiments may be required for different datasets and this strategy is not practical under the framework of Gaussian modeling of oriented object detection.
>
> **Q3**. KLD is invalid on horizontal detection.
>
> **A3**.  The horizontal box is a special case and still has the self-modulated mechanism analyzed in this paper. Although Eq. (4) and (16) are similar, there are two differences: more terms and different norms. **According to Tab. 6, the above differences make KLD better than baseline on AP indicators, especially AP75, comparable to GIoU.** It should also be noted that the gt for rotation detection is the minimum bounding rectangle, which contains as little background area as possible, and the horizontal special cases described in the paper also meet the above requirements. Although the gt of the COCO is horizontal box, it is not the minimum bounding rectangle, so the experimental results will be affected to a certain extent.
> As Reviewer eGtr said: "I do not consider this is a major issue as this works very well rotated object detection which is also valuable."
>
> **We hope this response could help address your concerns, and wish to receive your further feedback soon.**

---

> ### Author Response · Authors · 2021-08-31
> **Rebuttal to Reviewer MV1m**
>
> Could you please give us a reply to our rebuttal?
>
> We look forward to discussing your questions about this paper with you.

---

> > ### Comment · Reviewer_MV1m · 2021-09-01
> > **Feedback to the rebuttal**
> >
> > Thanks for your reply.
> >
> > I have read the rebuttal and some of my concerns have been addressed. However, I still think the good performance of  KLD loss is mainly because that KLD loss couples the rotation matrix $R(\theta)\in SO(2)$ with the offset predictions $\Delta x$ and $\Delta y$. That is the reason why the proposed approach can only address the rotated object detection.  Although the reviewer eGtr did not consider this limitation as a major issue, I still have concerns about it as the authors discussed the analysis of high-precision detection under the assumption of $\theta_t = 0$.
> >
> > Since the authors have addressed other concerns and I agree with the reviewer eGtr on the point for the strong results on rotated object detection, I would like to update my rating to 6.

---

> > > ### Author Response · Authors · 2021-09-01
> > > **KLD is not good enough on horizontal detection.**
> > >
> > > Thanks for your quick reply.
> > >
> > > Can we look at this issue from a different perspective: **The annotation defects (inaccuracy of the bounding box) of the dataset (e.g. COCO) caused the KLD to perform insufficiently, and it is not that the KLD is not good enough.**
> > >
> > > The ground truth for rotation detection is the minimum circumscribed rectangle, which means that ground truth can well reflect the true scale and direction information of the object. The "special horizontal case" discussed in this paper also meets the above requirements, the horizontal circumscribed rectangle is equal to the minimum circumscribed rectangle at this time.
> > >
> > > Although the ground truth of the COCO is a horizontal box, it is not the minimum circumscribed rectangle, which means that it loses the direction information and accurate scale information of the object. For example, a baseball bat placed obliquely in the image, the height and width of its horizontal circumscribed rectangle do not represent the height and width of the object itself. This causes that when KLD is applied to the COCO, the optimization mechanism of KLD that dynamically adjusts the angle gradient according to the aspect ratio is meaningless, which affects the improvement of the final performance.
> > >
> > > In fact, it is inappropriate to use the COCO to discuss $\theta=0$, because the COCO discards $\theta$ parameter. In addition, $\theta=0$ describes the instances in the horizontal position, but not mean all instances of the dataset are in a horizontal position. This paper uses COCO to discuss the "special horizontal case" to express that even if the dataset has certain labeling defects, KLD can have certain effects. After all, it is difficult to observe the performance improvement of all horizontal objects on the rotating dataset.
> > >
> > > Looking forward to your reply, thank you again for your improved rating.

---

### Decision · Program_Chairs · 2021-09-27

**Decision:**

Accept (Poster)

**Comment:**

The reviewers have discussed the paper and have not come to an agreement.
Reasons to accept: the proposed approach is noted to be simple and yet effective, and it can improve the performance of various existing object detectors on different datasets significantly. Most of the issues raised in the reviews were properly addressed in the author responses (including extra experimental results).
Reasons to reject: some of the reviewers were not satisfied with the provided responses and decided not to support the paper.

I think that the overall pros of the paper outweigh the cons and would like to recommend accepting the paper.